# Algorithms with Calibrated Machine Learning Predictions

**Judy Hanwen Shen** [1]   **Ellen Vitercik** [1 2]   **Anders Wikum** [2]

## Abstract

The field of *algorithms with predictions* incorporates machine learning advice in the design of online algorithms to improve real-world performance. A central consideration is the extent to which predictions can be trusted—while existing approaches often require users to specify an aggregate trust level, modern machine learning models can provide estimates of prediction-level uncertainty. In this paper, we propose *calibration* as a principled and practical tool to bridge this gap, demonstrating the benefits of calibrated advice through two case studies: the *ski rental* and *online job scheduling* problems. For ski rental, we design an algorithm that achieves near-optimal prediction-dependent performance and prove that, in high-variance settings, calibrated advice offers more effective guidance than alternative methods for uncertainty quantification. For job scheduling, we demonstrate that using a calibrated predictor leads to significant performance improvements over existing methods. Evaluations on real-world data validate our theoretical findings, highlighting the practical impact of calibration for algorithms with predictions.

## 1. Introduction

In recent years, advances in machine learning (ML) models have inspired researchers to revisit the design of classic online algorithms, incorporating insights from ML-based advice to improve decision-making in real-world environments. This research area, termed *algorithms with predictions*, seeks to design algorithms that are both robust to worst-case inputs and achieve performance that improves with prediction accuracy (a desideratum termed *consistency*) (Lykouris & Vassilvitskii, 2018). Many learning-augmented

algorithms have been developed for online decision-making tasks ranging from rent-or-buy problems like ski rental (Purohit et al., 2018; Anand et al., 2020; Sun et al., 2024) to sequencing problems like job scheduling (Cho et al., 2022).

This framework often produces a family of algorithms indexed by a single parameter intended to reflect the *global* reliability of the ML advice. Extreme settings of this parameter yield algorithms that make decisions as if the predictions are either all perfect or all uninformative (e.g., Mahdian et al., 2007; Lykouris & Vassilvitskii, 2018; Purohit et al., 2018; Rohatgi, 2020; Wei & Zhang, 2020; Antoniadis et al., 2020). In contrast, ML models often produce *local*, prediction-specific uncertainty estimates, exposing a disconnect between theory and practice. For instance, many neural networks provide calibrated probabilities or confidence intervals for each data point.

In this paper, we demonstrate that *calibration* can serve as a powerful tool to bridge this gap. An ML predictor is said to be calibrated if the probabilities it assigns to events match their observed frequencies; when the model outputs a high probability, the event is indeed likely, and when it assigns a low probability, the event rarely occurs. Calibrated predictors convey their uncertainty on each prediction, allowing decision-makers to safely rely on the model's advice, and eliminating the need for ad-hoc reliability estimates. Moreover, calibrating an ML model can easily be accomplished using popular methods (e.g. Platt Scaling (Platt et al., 1999) or Histogram Binning (Zadrozny & Elkan, 2001)) that reduce overconfidence (Vasilev & D'yakonov, 2023).

Although we are the first to study calibration for algorithms with predictions, Sun et al. (2024) proposed using *conformal prediction* in this setting—a common tool in uncertainty quantification (Vovk et al., 2005; Shafer & Vovk, 2008). Conformal predictions provide instance-specific confidence intervals that cover the target with high probability. While these approaches are orthogonal, we prove that calibration can offer key advantages over conformal prediction, especially when the predicted quantities have high variance. In extreme cases, conformal intervals can become too wide to be informative: for binary predictions, a conformal approach returns $\{0, 1\}$ unless the true label is nearly certain to be $0$ or $1$. In contrast, calibration still conveys information that aids decision-making.

---

[1]Department of Computer Science, Stanford University, Stanford, CA, USA [2]Department of Management Science & Engineering, Stanford University, Stanford, CA, USA. Correspondence to: Anders Wikum <wikum@stanford.edu>.

*Proceedings of the 42nd International Conference on Machine Learning*, Vancouver, Canada. PMLR 267, 2025. Copyright 2025 by the author(s).

## 1.1. Our contributions

We demonstrate the benefit of using calibrated predictors through two case studies: the ski rental and online job scheduling problems. Theoretically, we develop and give performance guarantees for algorithms that incorporate calibrated predictions. We validate our theoretical findings with strong empirical results on real-world data, highlighting the practical benefits of our approach.

**Ski rental.** The *ski rental problem* serves as a prototypical example of a broad family of online rent-or-buy problems, where one must choose between an inexpensive, short-term option (renting) and a more costly, long-term option (buying). In this problem, a skier will ski for an unknown number of days and, each day, must decide to either rent skis or pay a one-time cost to buy them. Generalizations of the ski rental problem have informed a broad array of practical applications in networking (Karlin et al., 2001), caching (Karlin et al., 1988), and cloud computing (Khanafer et al., 2013).

We design an online algorithm for ski rental that incorporates predictions from a calibrated predictor. We prove that our algorithm achieves optimal expected prediction-level performance for general distributions over instances and calibrated predictors. At a distribution level, its performance degrades smoothly as a function of the mean-squared error and calibration error of the predictor. Moreover, we demonstrate that calibrated predictions can be more informative than the conformal predictions of Sun et al. (2024) when the distribution over instances has high variance that is not explained by features, leading to better performance.

**Scheduling.** We next study online scheduling in a setting where each job has an urgency level, but only a machine-learned estimate of that urgency is available. This framework is motivated by scenarios such as medical diagnostics, where machine-learning tools can flag potentially urgent cases but cannot fully replace human experts.

We demonstrate that using a calibrated predictor provides significantly better guarantees than prior work (Cho et al., 2022), which approached this problem by ordering jobs based on the outputs of a binary predictor. We identify that this method implicitly relies on a crude form of calibration that assigns only two distinct values, resulting in many ties that must be broken randomly. In contrast, we prove that a properly calibrated predictor with finer-grained confidence levels provides a more nuanced job ordering, rigorously quantifying the resulting performance gains.

## 1.2. Related work

**Algorithms with predictions.** There has been significant recent interest in integrating ML advice into the design of online algorithms (see, e.g., Mitzenmacher & Vassilvitskii

(2022) for a survey). Much of the research provides a parameterized family of algorithms with no assumption on the reliability of predictions (e.g., Lykouris & Vassilvitskii, 2018; Purohit et al., 2018; Wei & Zhang, 2020). Subsequent work has studied more practical settings, such as assuming access to ML predictors learned from samples (Anand et al., 2020), with probabilistic correctness guarantees (Gupta et al., 2022), with a known confusion matrix (Cho et al., 2022), or that provide distributional predictions (Dinitz et al., 2024; Angelopoulos et al., 2024; Lin et al., 2022; Diakonikolas et al., 2021). While conceptually related, these papers do not study uncertainty quantification.

Recently, Sun et al. (2024) proposed a framework for quantifying prediction-level uncertainty based on conformal prediction. We show that calibration can offer key advantages over conformal prediction in this context, particularly when predicted quantities exhibit high variance.

**Calibration for decision-making.** A recent line of work examines calibration as a tool for downstream decision-making. Gopalan et al. (2023) show that a multi-calibrated predictor can be used to optimize any convex, Lipschitz loss function of an action and binary label. Zhao et al. (2021) adapt the required calibration guarantees to specific offline decision-making tasks, while Noarov et al. (2023) extend this algorithmic framework to the online adversarial setting. Though closely related to our work, these results do not extend to the (often unwieldy) loss functions encountered in competitive analysis.

## 2. Preliminaries

For clarity, we follow the convention that capital letters (e.g., $X$) denote random variables and lowercase letters denote realizations of random variables (e.g., the event $f(X) = v$).

**Learning-augmented algorithm design.** With each algorithmic task, we associate a set $\mathcal{I}$ of possible instances, a set $\mathcal{X}$ of features for those instances, and a joint distribution $\mathcal{D}$ over $\mathcal{X} \times \mathcal{I}$. Given a target function $T : \mathcal{I} \to \mathcal{Y}$ that provides information about each instance, we assume access to a predictor $f : \mathcal{X} \to \mathcal{Z} \supseteq \mathcal{Y}$ that has been trained to predict the target over $\mathcal{D}$. Let $R(f)$ denote the range of $f$.

If $\mathcal{A}(v, i)$ is the cost incurred by algorithm $\mathcal{A}$ with prediction $f(X) = v$ on instance $i \in \mathcal{I}$, and $\text{OPT}(i)$ is that of the offline optimal solution, the goal is to minimize either the *expected competitive ratio (CR)*

$$\mathop{\mathbb{E}}_{(X,I)\sim\mathcal{D}} \left[ \frac{\mathcal{A}(f(X), I)}{\text{OPT}(I)} \right]$$

or the *expected additive regret* $\mathbb{E}\left[\mathcal{A}(f(X), I) - \text{OPT}(I)\right]$, depending on context. Both measure the performance of $\mathcal{A}$ relative to OPT over $\mathcal{D}$. The former is consistent with

prior work on training predictors from samples for algorithms with predictions (Anand et al., 2020), while the latter is commonly used to quantify suboptimality in learning-augmented scheduling (Lindermayr & Megow, 2022; Im et al., 2023). When $\mathcal{D}$ and $f$ are clear from context, we refer to these quantities as $\mathbb{E}[\mathrm{CR}(\mathcal{A})]$ and $\mathbb{E}[\mathrm{R}(\mathcal{A})]$, respectively.

**Calibration.** An ML model is said to be *calibrated* if its predictions are, on average, correct. Formally,

**Definition 2.1.** A predictor $f : \mathcal{X} \to \mathcal{Z}$ with target $T : \mathcal{I} \to \mathcal{Y}$ is calibrated over $\mathcal{D}$ if

$$\mathop{\mathbb{E}}_{(X,I)\sim\mathcal{D}}[T(I) \mid f(X)] = f(X).$$

When $\mathcal{Y} = \{0, 1\}$, the equivalent condition $\Pr[T(I) = 1 \mid f(X)] = f(X)$ requires that $f(X)$ is a reliable probabilistic estimate of the event $\{T(I) = 1\}$.

A classic result from the literature on probabilistic forecasting states that calibrated predictions are the global minimizers of proper loss functions (DeGroot & Fienberg, 1983). However, achieving perfect calibration is difficult in practice. As a result, post-hoc calibration methods aim to minimize calibration error, such as the *max calibration error*, which measures the largest deviation from perfect calibration for any prediction.

**Definition 2.2.** The max calibration error of a predictor $f : \mathcal{X} \to \mathcal{Z}$ with target $T : \mathcal{I} \to \mathcal{Y}$ over $\mathcal{D}$ is

$$\max_{v \in R(f)} |v - \mathbb{E}[T(I) \mid f(X) = v]|.$$

Given any black box ML model and sufficient data, these methods yield a new predictor with a desired level of calibration error with high probability.

# 3. Ski Rental

In this section, we analyze calibration as a tool for uncertainty quantification in the classic online ski rental problem. All omitted proofs in this section are in Appendix A.

## 3.1. Setup

**Problem.** A skier plans to ski for an unknown number of days $Z \in \mathbb{N}$ and has two options: buy skis at a one-time cost of $b \in \mathbb{N}$ dollars or rent them for 1 dollar per day. The goal is to determine how many days to rent before buying, minimizing the total cost. If $Z = z$ were known *a priori*, the optimal policy would rent for $b$ days when $z < b$ and buy immediately otherwise, costing $\min\{z, b\}$. Without knowledge of $z$, competitive ratios of 2 (Karlin et al., 1988) and $\frac{e}{e-1}$ (Karlin et al., 1994) are tight for deterministic and random strategies, respectively. For convenience, we study a continuous variant of this problem where $Z, b, k \in \mathbb{R}_{\geq 0}$ as in prior work (Anand et al., 2020; Sun et al., 2024).

---

**Algorithm 1** $\mathcal{A}_{k_*}$

---

**input:** prediction $f(X) = v$, max calibration error $\alpha$
**if** $v \leq \frac{4+3\alpha}{5}$ **then**
    Rent for $b$ days before buying.
**else**
    Rent for $b\sqrt{\frac{1-v+\alpha}{v+\alpha}}$ days before buying.
**end if**

---

**Predictions.** Let $\mathcal{X}$ be a set of skier features, $\mathcal{I} = \mathbb{R}_{\geq 0}$ be the set of possible days skied, and $\mathcal{D}$ be an unknown distribution over feature/duration pairs $\mathcal{X} \times \mathbb{R}_{\geq 0}$. Motivated by the form of the optimal offline algorithm, we analyze a calibrated predictor $f : \mathcal{X} \to [0, 1]$ for the target $T(z) = \mathbb{1}_{\{z > b\}}$, indicating if the skier will ski for more than $b$ days. For $(X, Z) \sim \mathcal{D}$, a prediction of $f(X) \approx 1$ (respectively, $f(X) \approx 0$) means $Z > b$ (respectively, $Z \leq b$) with high certainty.

**Learning-augmented ski rental.** A deterministic learning-augmented algorithm $\mathcal{A}_k$ for ski rental takes as input a prediction $f(X) = v$ and returns a recommendation: "rent skis for $k(v)$ days before buying." The cost of following this policy when skiing for $z$ days is

$$\mathcal{A}_k(v, z) = \begin{cases} k(v) + b & \text{if } z > k(v) \\ z & \text{if } z \leq k(v) \end{cases}.$$

We aim to select $k : [0, 1] \to \mathbb{R}_+$ to minimize $\mathbb{E}[\mathrm{CR}(\mathcal{A}_k)]$.

## 3.2. Ski rental with calibrated predictions

In Algorithm 1, we introduce a deterministic policy for ski rental based on calibrated predictions. To avoid following bad advice, the algorithm defaults to a worst-case strategy of renting for $b$ days unless sufficiently confident that the skier will ski for at least $b$ days. In this second case, the algorithm smoothly interpolates between a strategy that rents for $b\sqrt{(1-\alpha)/\alpha}$ days and one that rents for $b\sqrt{\alpha/(1+\alpha)}$ days, where $\alpha \in [0, 1]$ is a bound on local calibration error that hedges against greedily following predictions.

**Theorem 3.1.** *Given a predictor $f$ with mean-squared error $\eta$ and max calibration error $\alpha$, Algorithm 1 achieves $\mathbb{E}[\mathrm{CR}(\mathcal{A}_{k_*})] \leq 1 + 2\alpha + \min\left\{\mathbb{E}[f(X)] + \alpha, 2\sqrt{\eta + 3\alpha}\right\}.$*

As the predictor becomes more accurate (i.e., both $\eta$ and $\alpha$ decrease), the algorithm's expected CR approaches 1. The rest of this subsection will build to a proof of Theorem 3.1.

**Prediction-level analysis.** We begin by upper bounding $\mathbb{E}[\mathrm{CR}(\mathcal{A}_k) \mid f(X) = v]$. Let $B_v = \{f(X) = v\}$ be the event that $f$ predicts $v \in R(f)$ and $C = \{Z > b\}$ be the

*Table 1.* Objective values for fixed prediction $f(X) = v$, $z$ days skied, and renting for $k(v)$ days.

| CONDITION | OPT($z$) | $\mathcal{A}_k(v, z)$ |
|---|---|---|
| (i)   $z \leq \min\{k(v),\ b\}$ | $z$ | $z$ |
| (ii)  $k(v) < z \leq b$ | $z$ | $k(v) + b$ |
| (iii) $b < z \leq k(v)$ | $b$ | $z$ |
| (iv)  $z > \max\{k(v),\ b\}$ | $b$ | $k(v) + b$ |

event that the number of days skied is more than $b$. Then

$$\mathbb{E}[\mathrm{CR}(\mathcal{A}_k) \mid B_v] = \mathbb{E}[\mathrm{CR}(\mathcal{A}_k) \mid B_v, C] \cdot \Pr[C \mid B_v] \quad (1)$$
$$+ \mathbb{E}[\mathrm{CR}(\mathcal{A}_k) \mid B_v, C^c] \cdot \Pr[C^c \mid B_v].$$

Lemma 3.2 bounds each of the quantities from Equation (1).

**Lemma 3.2.** *Given a predictor $f$ with max calibration error $\alpha$, for all $v \in R(f)$,*

1. $\Pr[C \mid f(X) = v] \leq v + \alpha$

2. $\Pr[C^c \mid f(X) = v] \leq 1 - v + \alpha$

3. $\mathbb{E}[\mathrm{CR}(\mathcal{A}_k) \mid B_v, C] \leq 1 + \frac{k(v)}{b}$

4. $\mathbb{E}[\mathrm{CR}(\mathcal{A}_k) \mid B_v, C^c] \leq 1 + \frac{b \cdot \mathbb{1}_{\{k(v) < b\}}}{k(v)}$.

*Proof sketch.* (1) and (2) follow from the fact that $f$ predicts $\mathbb{1}_C$ with max calibration error $\alpha$. Under $C = \{Z \geq b\}$, one of conditions (iii) or (iv) from Table 3 hold. In either case, $\mathcal{A}_k(v, Z)/\mathrm{OPT}(Z) \leq 1 + \frac{k(v)}{b}$. Under $C^c$, one of conditions (i) or (ii) hold. $\mathrm{CR}(\mathcal{A}_k) = 1$ for (i). For (ii),

$$\frac{\mathcal{A}_k(v, Z)}{\mathrm{OPT}(Z)} \leq \frac{k(v) + b}{k(v)} = 1 + \frac{b \cdot \mathbb{1}_{\{k(v) < b\}}}{k(v)}.$$

$\square$

Applying all four bounds to Equation (1) yields

$$\mathbb{E}[\mathrm{CR}(\mathcal{A}_k) \mid f(X) = v] \leq \qquad\qquad (2)$$
$$1 + 2\alpha + \frac{(v + \alpha)k(v)}{b} + \mathbb{1}_{\{k(v) < b\}} \cdot \frac{(1 - v + \alpha)b}{k(v)}.$$

The renting strategy $k_*(v)$ from Algorithm 1 is the minimizer of the upper bound in Equation (2).

**Theorem 3.3.** *Given a predictor $f$ with max calibration error $\alpha$, for any prediction $v \in R(f)$, Algorithm 1 achieves*

$$\mathbb{E}[\mathrm{CR}(\mathcal{A}_{k_*}) \mid f(X) = v] \leq$$
$$1 + 2\alpha + \min\{v + \alpha, 2\sqrt{(v + \alpha)(1 - v + \alpha)}\}.$$

*Proof sketch.* Given a prediction $f(X) = v$, Algorithm 1 rents for $k_*(v)$ days where

$$k_*(v) = \begin{cases} b & \text{if } 0 \leq v \leq \frac{4 + 3\alpha}{5} \\ b\sqrt{\frac{1 - v + \alpha}{v + \alpha}} & \text{if } \frac{4 + 3\alpha}{5} < v \leq 1. \end{cases}$$

Evaluating the right-hand-side of Equation (2) at $k_*(v)$ gives

$$\begin{cases} 1 + 2\alpha + (v + \alpha) & \text{if } 0 \leq v \leq \frac{4 + 3\alpha}{5} \\ 1 + 2\alpha + 2\sqrt{(v + \alpha)(1 - v + \alpha)} & \text{if } \frac{4 + 3\alpha}{5} < v \leq 1. \end{cases}$$

The fact that $v + \alpha \leq 2\sqrt{(v + \alpha)(1 - v + \alpha)}$ for $v \in [0, \frac{4 + 3\alpha}{5}]$ and $v + \alpha > 2\sqrt{(v + \alpha)(1 - v + \alpha)}$ for $v \in (\frac{4 + 3\alpha}{5}, 1]$ completes the proof. $\square$

Moreover, no deterministic learning-augmented algorithm for ski rental can outperform Algorithm 1 for general distributions $\mathcal{D}$ and calibrated predictors $f$. The construction is non-trivial, so we refer the reader to the proof in Appendix A.

**Theorem 3.4.** *For all renting strategies $k : [0, 1] \to \mathbb{R}_+$, predictions $v \in [0, 1]$ and $\epsilon > 0$, there exists a distribution $\mathcal{D}_v^\epsilon$ and a calibrated predictor $f$ such that*

$$\mathbb{E}[\mathrm{CR}(\mathcal{A}_k) \mid f(X) = v] \geq 1 + \min\left\{v, 2\sqrt{v(1 - v)}\right\} - \epsilon.$$

**Global analysis.** In extracting a global bound from the conditional guarantee in Theorem 3.3, we encounter a term $(f(X) + \alpha)(1 - f(X) + \alpha)$ that is an upper bound on the variance of the conditional distribution $\mathbb{1}_{\{Z \geq b\}} \mid f(X)$. Lemma 3.5 relates this quantity to error statistics of $f$.

**Lemma 3.5.** *If $f : \mathcal{X} \to [0, 1]$ has mean-squared error $\eta$ and max calibration error $\alpha$, then*

$$\mathbb{E}[f(X)(1 - f(X))] \leq \eta + \alpha.$$

Finally, we prove this section's main theorem.

*Proof of Theorem 3.1.* By the tower property of conditional expectation, $\mathbb{E}[\mathrm{CR}(\mathcal{A}_{k_*})] = \mathbb{E}\big[\mathbb{E}[\mathrm{CR}(\mathcal{A}_{k_*}) \mid f(X)]\big]$. Applying Theorem 3.3 yields

$$\mathbb{E}[\mathrm{CR}(\mathcal{A}_{k_*})] \leq 1 + 2\alpha$$
$$+ \mathbb{E}\left[\min\{f(X) + \alpha, 2\sqrt{(f(X) + \alpha)(1 - f(X) + \alpha)}\}\right].$$

Recall that $\mathbb{E}[\min(X, Y)] \leq \min(\mathbb{E}[X], \mathbb{E}[Y])$ for random variables $X, Y$. Furthermore, the function $h(y) = \sqrt{(y + \alpha)(1 - y + \alpha)}$ is concave over the unit interval, so by Jensen's inequality

$$\mathbb{E}\left[\min\{f(X) + \alpha, 2\sqrt{(f(X) + \alpha)(1 - f(X) + \alpha)}\}\right] \leq$$
$$\min\{\mathbb{E}[f(X)] + \alpha, 2\sqrt{\mathbb{E}[(f(X) + \alpha)(1 - f(X) + \alpha)]}\}.$$

Finally, observe that

$$(f(X) + \alpha)(1 - f(X) + \alpha) \leq f(X)(1 - f(X)) + 2\alpha.$$

We apply Lemma 3.5 to bound $\mathbb{E}[f(X)(1 - f(X))]$. $\square$

### 3.3. Comparison to previous work

**Consistency and robustness.** It is well known that for $\lambda \in (0, 1)$, any $(1 + \lambda)$-consistent algorithm for deterministic ski rental must be at least $(1 + \frac{1}{\lambda})$-robust (Wei & Zhang, 2020; Angelopoulos et al., 2020; Gollapudi & Panigrahi, 2019). While Algorithm 1 is subject to this trade-off in the worst case, calibration provides sufficient information to hedge against adversarial inputs in expectation, leading to substantial improvements in average-case performance. Indeed, it can be seen from the bound in Theorem 3.3 that Algorithm 1 is 1-consistent and always satisfies $\mathbb{E}[\mathrm{CR}(\mathcal{A}_{k_*})] \leq 1.8$ when advice is calibrated ($\alpha = 0$). An analysis similar to that of Theorem 15 in Anand et al. (2020) shows that Algorithm 1 is $g(\alpha)$-robust, where

$$g(\alpha) = \begin{cases} 1 + \sqrt{\frac{1+\alpha}{\alpha}} & \text{if } \alpha < 1/3 \\ 2 & \text{if } \alpha \geq 1/3 \end{cases}$$

is a decreasing function of $\alpha$. This is because Algorithm 1 executes a worst-case 2-competitive strategy when $\alpha \geq 1/3$ and never buys skis before day $b\sqrt{\frac{\alpha}{1+\alpha}}$ otherwise.

We note that one can run the same algorithm using an artificial upper bound $\alpha' > \alpha$ on max calibration error to achieve an improved robustness level $g(\alpha')$. As seen from the bounds in Theorem 3.3 and Theorem 3.1, this adjustment will come at the cost of expected performance, highlighting the tradeoff between average and worst-case performance.

**Uncertainty quantification.** We are not the first to explore uncertainty quantified predictions for ski rental. Sun et al. (2024) take an orthogonal approach based on conformal prediction. Their method, Algorithm 2, assumes access to a probabilistic interval predictor $\mathrm{PIP}_\delta : \mathcal{X} \to \mathcal{P}([0, 1])$. $\mathrm{PIP}_\delta$ outputs an interval $[\ell, u] = \mathrm{PIP}_\delta(X)$ containing the true number of days skied $Z \in [\ell, u]$ with probability at least $1 - \delta$. Interval predictions are especially useful when the uncertainty $\delta$ and interval width $u - \ell$ are both small.

However, as features become less informative, the width of prediction intervals must increase to maintain the same confidence level. This can result in intervals that are too wide to provide meaningful insight into the true number of days skied. Lemma 3.6 and Theorem 3.7 demonstrate that there are infinite families of distributions for which calibrated predictions are more informative than conformal predictions for ski rental.

**Lemma 3.6.** *For all $a \in [0, 1/2]$, there exists an infinite family of input distributions for which Algorithm 2 defaults to a worst-case break-even strategy for all interval predictors $\mathrm{PIP}_\delta$ with uncertainty $\delta < a$.*

*Proof sketch.* The construction places mass $1 - a$ on some day $z_1 \leq \frac{b}{2}$ and mass $a$ on $z_2 \geq 2b$. Any $\mathrm{PIP}_\delta$ with $\delta < a$

---

**Algorithm 2** (Sun et al., 2024) Optimal ski rental with conformal predictions

**input:** interval prediction $[\ell, u] = \mathrm{PIP}_\delta(X)$
**if** $\ell \leq u < b$ **then**
    Rent for $b$ days
**else if** $b < \ell \leq u$ **then**
    Rent for $b \cdot \min\{\sqrt{\delta/1 - \delta}, 1\}$ days
**else**
    **if** $\zeta(\delta, \ell) \geq 2$ and $\delta + \frac{u}{b} \geq 2$ **then**
        Rent for $b$ days
    **else if** $\zeta(\delta, \ell) \leq \delta + \frac{u}{b}$ **then**
        Rent for $\ell \cdot \min\{\sqrt{b\delta/\ell(1 - \delta)}, 1\}$ days
    **else**
        Rent for $u$ days
    **end if**
**end if**

$$\zeta(\delta, \ell) := \begin{cases} \delta + \frac{(1-\delta)b}{\ell} + 2\sqrt{\frac{\delta(1-\delta)b}{\ell}} & \text{if } \delta \in [0, \frac{\ell}{\ell+b}) \\ 1 + \frac{b}{\ell} & \text{if } \delta \in [\frac{\ell}{\ell+b}, 1] \end{cases}$$

must output an interval $[\ell, u]$ containing both $z_1$ and $z_2$. Moreover, $\zeta(\delta, \ell) \geq 2$ and $\delta + \frac{u}{b} \geq 2$ by construction. $\square$

**Theorem 3.7.** *For all $a \in [0, 1/2]$, all instantiations $\mathcal{A}$ of Algorithm 2 using PIPs with uncertainty $\delta < a$, and all distributions from Lemma 3.6, if $f$ is a predictor with mean-squared error $\eta$ and max calibration error $\alpha$ satisfying $2\alpha + 2\sqrt{\eta + 3\alpha} < a$, then $\mathbb{E}[\mathrm{CR}(\mathcal{A}_{k_*})] < \mathbb{E}[\mathrm{CR}(\mathcal{A})]$.*

*Proof sketch.* For the distributions in Lemma 3.6, the number of days skied is greater than $b$ with probability $a$. Thus, the expected competitive ratio of the break-even strategy is $\mathbb{E}[\mathrm{CR}(\mathcal{A})] = a \cdot 2 + (1 - a) \cdot 1 = 1 + a$. The result follows from the bound on $\mathbb{E}[\mathrm{CR}(\mathcal{A}_{k_*})]$ given in Theorem 3.1. $\square$

## 4. Online Job Scheduling

In this section, we explore the role of calibration in a model for *scheduling with predictions* first proposed by Cho et al. (2022) to direct human review of ML-flagged abnormalities in diagnostic radiology. Omitted proofs from this section can be found in Appendix B.

### 4.1. Setup

**Problem.** There is a single machine (lab tech) that needs to process $n$ jobs (diagnostic images), each requiring one unit of processing time. Job $i$ has some unknown priority $y_i \in \{0, 1\}$ that is independently high ($y_i = 1$) with probability $\rho$ and low ($y_i = 0$) with probability $1 - \rho$. Although job priorities are unknown a priori, the priority $y_i$ is revealed after completing some fixed fraction $\theta \in (0, 1)$ of job $i$. Upon learning $y_i$, a scheduling algorithm can choose

to complete job $i$, or switch to a new job and "store" job $i$ for completion at a later time. The goal is to schedule the $n$ jobs in a way that minimizes the weighted sum of completion times $\sum_{i=1}^n C_i \cdot \omega_{y_i}$ where $C_i$ is the completion time of job $i$, and $\omega_1 > \omega_0 > 0$ are costs associated with delaying a job of each priority for one unit of time. In hindsight, it is optimal to schedule jobs in decreasing order of priority.

**ML predictions.** Based on the assumption that the $n$ jobs to be scheduled are iid, let $\mathcal{X} = \mathcal{X}_0^n$ be a set of job features, $\mathcal{I} = \{0,1\}^n$ be the set of possible priorities, and $\mathcal{D} = \mathcal{D}_0^n$ be an unknown joint distribution over feature/priority pairs. The prediction task for this problem involves training a predictor $f$ whose target is the true priority of each job $T(\vec{y}) = \vec{y}$. This amounts to training a 1-dimensional predictor $f : \mathcal{X}_0 \to \mathcal{Z}$ that acts on the $n$ jobs independently: $f(\vec{X}) := (f(\vec{X}_1), \ldots, f(\vec{X}_n))$.

**Learning-augmented scheduling.** Cho et al. (2022) introduce a threshold-based scheduling rule informed by probabilities $p_i$ that job $i$ is high priority based on identifying features (Algorithm 3). Their algorithm switches between two extremes—a *preemptive* policy that starts a new job whenever the current job is revealed to be low priority, and a *non-preemptive* policy that completes any job once it is begun—based on the threshold parameter

$$\beta := \frac{\theta}{1-\theta} \cdot \frac{\omega_1}{\omega_1 - \omega_0}.$$

In detail, jobs are opened in decreasing order of $p_i$. Jobs with $p_i > \beta$ are processed preemptively, and the remaining jobs are processed non-preemptively.

A learning-augmented algorithm $\mathcal{A}$ for job scheduling determines the probabilities $p_i$ from ML advice. Cho et al. (2022) assume access to a binary predictor $f_b : \mathcal{X}_0 \to \{0,1\}$ of job priority and study the case where $p_i = \Pr[\vec{Y}_i = 1 \mid f_b(\vec{X}_i)]$. These probabilities can be computed using Bayes' rule, and because $f_b$ is binary, this procedure effectively assigns each job one of two probabilities. Although not explicitly discussed by Cho et al. (2022), this amounts to a basic form of post-hoc calibration. In contrast, our results extend to arbitrary calibrated predictors $f : \mathcal{X}_0 \to [0,1]$—a more general framework that calls for new mathematical techniques—allowing us to significantly improve upon their results. In this setting, $\mathcal{A}$ takes the predictions $f(\vec{X}) = \vec{v}$ as input and executes Algorithm 3 with probabilities $p_i = \vec{v}_i$.

To quantify the optimality gap of $\mathcal{A}$, Cho et al. (2022) note that compared to OPT, Algorithm 3 incurs (1) a cost of $\theta\omega_1$ for each *inversion*, or pair of jobs whose true priorities $y_i$ are out of order, and (2) a cost of $\theta\omega_0$ for each pair of low priority jobs encountered when acting preemptively. When acting non-preemptively, Algorithm 3 incurs (3) a cost of $\omega_1 - \omega_0$ for each inversion. Thus, for fixed predictions

**input:** Probabilities $\{p_i\}_{i=1}^n$ that each job is high-priority
Define $n_1 = |\{i : p_i > \beta\}|$
Order probabilities $p_{(1)} \geq \cdots \geq p_{(n)}$
Run jobs $j_{(1)}, \ldots, j_{(n_1)}$ preemptively, in order
Complete remaining jobs non-preemptively, in order

---

$f(\vec{X}) = \vec{v}$ and true job priorities $\vec{y}$,

$$\mathcal{A}(\vec{v}, \vec{y}) - \text{OPT}(\vec{y}) \tag{3}$$
$$= \theta\omega_1 L(\vec{v}, \vec{y}) + \theta\omega_0 M(\vec{v}, \vec{y}) + (\omega_1 - \omega_0)N(\vec{v}, \vec{y}),$$

where $L(\vec{v}, \vec{y})$, $M(\vec{v}, \vec{y})$, and $N(\vec{v}, \vec{y})$ count occurrences of (1), (2), and (3), respectively (see Table 2 for details).

### 4.2. Scheduling with calibrated predictions

**Calibration and job sequencing.** To build intuition for why finer-grained calibrated predictors sequence jobs more accurately, we begin by observing that Algorithm 3 orders jobs with the same probability $p_i$ randomly. Given a calibrated predictor $f$, consider the coarse calibrated predictor

$$f'(x) = \begin{cases} \mathbb{E}[f(X) \mid f(X) > \beta] & \text{if } f(x) > \beta \\ \mathbb{E}[f(X) \mid f(X) \leq \beta] & \text{if } f(x) \leq \beta \end{cases}$$

obtained by averaging the predictions of $f$ above and below the threshold $\beta$. Whereas $|R(f)|$ may be large, $f'$ is only capable of outputting $|R(f')| = 2$ values. As a result, when ordering jobs with features $X_1, \ldots, X_n$ according to predictions from $f'$, all jobs with $f(X) > \beta$ will be sequenced before jobs with $f(X) \leq \beta$, but the ordering of jobs within these bins will be random. In contrast, predictions from $f$ provide a more informative ordering of jobs (Figure 1). Note, however, that $f = f'$ when $f$ has no variance in its predictions above or below the threshold $\beta$. We demonstrate in Theorem 4.3 that this intuition holds in general: improvements scale with the granularity of predictions.

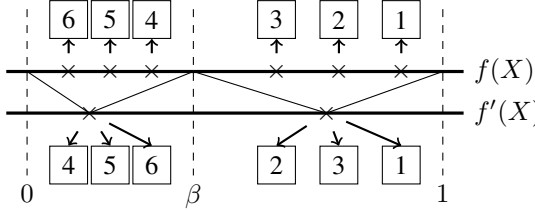

*Figure 1.* Job sequencing under fine-grained (above) and coarse (below) calibrated predictors. For six example jobs, predicted probabilities $p_i$ are marked with $\times$, and numbered boxes give the order of jobs according to each predictor.

Table 2. Quantities of interest in learning-augmented scheduling for fixed predictions $f(\vec{X}) = \vec{v}$ and job priorities $\vec{y}$.

| Quantity | Description | Relevant setting |
|---|---|---|
| $n_1 = |\{i : \vec{v}_i > \beta\}|$ | Number of jobs likely to be high priority. | — |
| $L(\vec{v}, \vec{y}) = \sum_{i=1}^{n_1} \sum_{j=i+1}^{n_1} \mathbb{1}_{\{\vec{y}_{(i)}=0 \wedge \vec{y}_{(j)}=1\}}$ | Number of inversions among jobs likely to be high priority. | Preemptive |
| $M(\vec{v}, \vec{y}) = \sum_{i=1}^{n_1} \sum_{j=i+1}^{n_1} \mathbb{1}_{\{\vec{y}_{(i)}=0 \wedge \vec{y}_{(j)}=0\}}$ | Number of low-priority job pairs among jobs likely to be high priority. | Preemptive |
| $N(\vec{v}, \vec{y}) = \sum_{i=1}^{n} \sum_{j=i+1}^{n} \mathbb{1}_{\{\vec{y}_{(i)}=0 \wedge \vec{y}_{(j)}=1\}} - L(\vec{v}, \vec{y})$ | Number of inversions among job pairs where at least one is likely to be low priority. | Non-preemptive |

**Performance analysis.** Building off of Equation (3), we bound the expected competitive ratio $\mathbb{E}[\mathrm{CR}(\mathcal{A})]$ by bounding each of $\mathbb{E}[L(f(\vec{X}), \vec{Y})]$, $\mathbb{E}[M(f(\vec{X}), \vec{Y})]$, and $\mathbb{E}[N(f(\vec{X}), \vec{Y})]$. The dependence on the ordering of predictions from $f$ in these random counts means our analysis heavily involves functions of order statistics. For example, considering the shared summand of $L(\cdot)$ and $N(\cdot)$,

$$\mathbb{E}\left[ \mathbb{1}_{\{\vec{Y}_{(i)}=0\}} \cdot \mathbb{1}_{\{\vec{Y}_{(j)}=1\}} \mid f(\vec{X}) \right]$$
$$= \left( \Pr[\vec{Y}_{(i)} = 0 \mid f(\vec{X}_{(i)})] \cdot \Pr[\vec{Y}_{(j)} = 0 \mid f(\vec{X}_{(j)})] \right)$$
$$= (1 - f(\vec{X}_{(i)})) f(\vec{X}_{(j)})$$
$$= g(f(\vec{X}_{(i)}), f(\vec{X}_{(j)}))$$

for the function $g(x, y) = (1-x)y$. Similarly, the analysis for the summand of $M(\cdot)$ yields $g(f(\vec{X}_{(i)}), f(\vec{X}_{(j)}))$ for $g(x, y) = (1-x)(1-y)$. Based on this, our high-level strategy is to relate "ordered" expectations of the form

$$\mathbb{E}\left[ \sum_{i=1}^{n} \sum_{j=i+1}^{n} g\big(f(\vec{X}_{(i)}), f(\vec{X}_{(j)})\big) \right]$$

to their "unordered" counterparts

$$\mathbb{E}\left[ \sum_{i=1}^{n} \sum_{j=i+1}^{n} g\big(f(\vec{X}_i), f(\vec{X}_j)\big), \right]$$

which are simple to compute. Lemma 4.1 shows that the ordered and unordered expectations are, in fact, equivalent when the function $g$ satisfies $g(x, y) = g(y, x)$.

**Lemma 4.1.** *Let $X_1, \ldots, X_n$ be iid random variables with order statistics $X_{(1)} \geq \cdots \geq X_{(n)}$. For any symmetric function $g : \mathbb{R} \times \mathbb{R} \to \mathbb{R}$,*

$$\sum_{i=1}^{n} \sum_{j=i+1}^{n} g(X_{(i)}, X_{(j)}) = \sum_{i=1}^{n} \sum_{j=i+1}^{n} g(X_i, X_j).$$

This result is sufficient to compute the expectation of $M(\cdot)$ exactly. For the other counts, the analysis is more technical as $g(x, y) = (1-x)y$ is not symmetric. Lemma 4.2 characterizes the relationship between the ordered and unordered expectations for the function $g(x, y) = (1-x)y$.

**Lemma 4.2.** *Let $X_1, \ldots, X_n$ be iid samples from a distribution over the unit interval $[0, 1]$ with order statistics $X_{(1)} \geq \cdots \geq X_{(n)}$. Then,*

$$\mathbb{E}\left[ \sum_{i=1}^{n} \sum_{j=i+1}^{n} (1 - X_{(i)}) \cdot X_{(j)} \right] \leq$$
$$\mathbb{E}\left[ \sum_{i=1}^{n} \sum_{j=i+1}^{n} (1 - X_i) \cdot X_j \right] - \binom{n}{2} \cdot \mathrm{Var}(X_1).$$

*Proof sketch.* By Lemma 4.1 with $g(x, y) = xy$,

$$\sum_{i=1}^{n} \sum_{j=i+1}^{n} X_{(i)} \cdot X_{(j)} = \sum_{i=1}^{n} \sum_{j=i+1}^{n} X_i \cdot X_j$$

can be removed from both sides. Then, we apply Lemma 4.1 with $g(x, y) = \min(x, y)$ to simplify the left-hand-side.

$$\sum_{i=1}^{n} \sum_{j=i+1}^{n} X_{(j)} = \sum_{i=1}^{n} \sum_{j=i+1}^{n} \min\{X_{(i)}, X_{(j)}\}$$
$$= \sum_{i=1}^{n} \sum_{j=i+1}^{n} \min\{X_i, X_j\}.$$

Finally, we show that $\mathbb{E}[X_1 - \min\{X_1, X_2\}] \geq \mathrm{Var}(X_1)$. Note that $\mathbb{E}[X_1] - \mathbb{E}[\min\{X_1, X_2\}] = \frac{1}{2}\mathbb{E}|X_1 - X_2|$ since

$$X_1 - \min\{X_1, X_2\} = \begin{cases} 0 & \text{if } X_1 \leq X_2 \\ |X_1 - X_2| & \text{if } X_1 > X_2. \end{cases}$$

Finally, $\mathbb{E}|X_1 - X_2| \geq \mathbb{E}|X_1 - X_2|^2 = 2\mathrm{Var}(X_1)$. $\quad\square$

With careful conditioning to deal with random summation bounds, we apply Lemma 4.2 to bound the expectations of

$L(\cdot)$ and $N(\cdot)$, giving this section's main theorem. Of note, Theorem 4.3 says that the expected number of inversions of high and low priority jobs decreases with predictor granularity, measured by $\kappa_1$ and $\kappa_2$. For the method from Cho et al. (2022), $\kappa_1 = \kappa_2 = 0$ and the inequalities hold with equality.

**Theorem 4.3.** *Let $f$ be calibrated, with $\Pr[f(X) > \beta \mid Y = 0] = \epsilon_0$, $\Pr[f(X) \le \beta \mid Y = 1] = \epsilon_1$,*

$$\kappa_1 = \Pr[f(X) > \beta]^2 \cdot \mathrm{Var}(f(X) \mid f(X) > \beta), \text{ and}$$
$$\kappa_2 = \Pr[f(X) \le \beta]^2 \cdot \mathrm{Var}(f(X) \mid f(X) \le \beta).$$

*Then*

1. $\mathbb{E}[L(f(\vec{X}), \vec{Y})] \le \binom{n}{2}\big(\rho(1-\rho)(1+\epsilon_0)\epsilon_1 - \kappa_1\big)$
2. $\mathbb{E}[M(f(\vec{X}), \vec{Y})] = \binom{n}{2}(1-\rho)^2\epsilon_0^2$
3. $\mathbb{E}[N(f(\vec{X}), \vec{Y})] \le \binom{n}{2}\big(\rho(1-\rho)\epsilon_0(1-\epsilon_1) - \kappa_2\big)$

*Remark* 4.4. $\mathcal{A}(f(\vec{X}), \cdot) - \mathrm{OPT}(\cdot) = 0$ when $\epsilon_0 = \epsilon_1 = 0$, and $\mathcal{A}$ inherits the robustness guarantees of Cho et al. (2022) when $\epsilon_0$ and $\epsilon_1$ are large.

An analogous result holds under the weaker assumption that $f$ monotonically calibrated. That is, the empirical frequencies $\Pr[Y = 1 \mid f(X)]$ are non-decreasing in the prediction $f(X)$. This property holds trivially for calibrated predictors, but zero calibration error is not required. In fact, many calibration approaches used in practice (e.g. Platt scaling (Platt et al., 1999) and isotonic regression (Zadrozny & Elkan, 2001)) produce a monotonically calibrated predictor with non-zero calibration error. See Appendix B for details.

## 5. Experiments

We now evaluate our algorithms on two real-world datasets, demonstrating the utility of using calibrated predictions. See Appendix C for additional details about our datasets and model training, as well a broader collection of results for different ML models and parameter settings.[1]

### 5.1. Ski rental: Citi Bike rentals

To model the rent-or-buy scenario in the ski rental problem, we use publicly available Citi Bike usage data.[2] This dataset has been used for forecasting (Wang, 2016), system balancing (O'Mahony & Shmoys, 2015), and transportation policy (Lei & Ozbay, 2021), but to the best of our knowledge, this is its first use for ski rental. In this context, a Citi Bike user can choose one of two options: pay by ride duration (rent) or purchase a day pass (buy). If the user plans

---

[1]Code and data available here: https://github.com/heyyjudes/algs-cali-pred

[2]Monthly usage data is publicly available at https://citibikenyc.com/system-data.

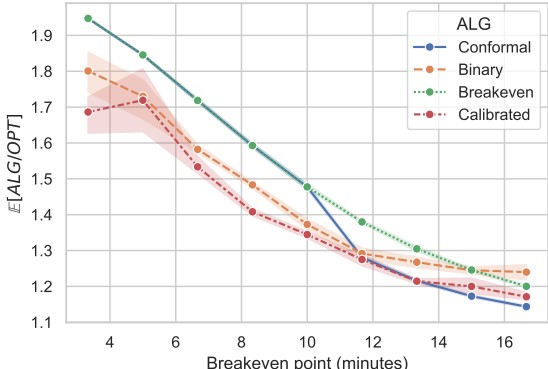

*Figure 2.* Comparison of $\mathbb{E}[\mathrm{ALG/OPT}]$ for algorithms aided by predictions from a small MLP with two hidden layers of size 8 and 2. Algorithm 1 (CALIBRATED) performs best on average.

to ride for longer than the break-even point of $b$ minutes, it is cheaper to buy a day pass than to pay by trip duration.[3] We use single-ride durations to approximate the rent vs. buy trade-off for a spectrum of break-even points $b$. The distribution over ride durations can be seen in Appendix C.

We analyze the impact of advice from multiple predictor families, including XGBoost, logistic regression, and small multi-layer perceptrons (MLP). Each predictor has access to available ride features: start time, start location, user age, user gender, user membership, and approximate end station latitude. While these features are not extremely informative, most predictor families are able to achieve AUC and accuracy above 0.8 for $b > 6$. Figure 2 summarizes the expected competitive ratios achieved by our method from Algorithm 1 (CALIBRATED) and baselines from previous work when given advice from a small neural network. Baselines include the worst-case optimal deterministic algorithm that rents for $b$ minutes (Karlin et al., 1988) (BREAKEVEN), the black-box binary predictor ski-rental algorithm by Anand et al. (2020) (BINARY), and the PIP algorithm described in Algorithm 2 (Sun et al., 2024) (CONFORMAL). Though each algorithm is aided by predictors from the same family, the actual advice may differ. For example, CONFORMAL assumes access to a regressor that predicts ride duration directly. While performance is distribution-dependent, we see that our calibration-based approach often leads to the most cost-effective rent/buy policy in this scenario.

### 5.2. Scheduling: sepsis triage

We use a real-world dataset for sepsis prediction to validate our theory results for scheduling with calibrated predictions. Sepsis is a life-threatening response to infection that typically appears after hospital admission (Singer et al., 2016).

---

[3]The day pass is designed to be more economical for multiple unlocks of a bike (e.g., $b \approx 66$ minutes for 1 unlock). However, ride data is anonymous, so we cannot track daily usage.

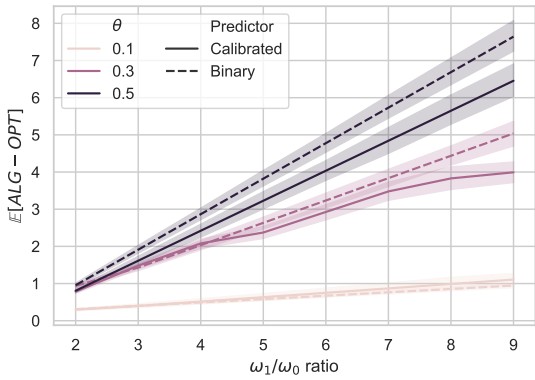

*Figure 3.* Comparison of $\mathbb{E}[\mathrm{ALG} - \mathrm{OPT}]$ (normalized) achieved by Algorithm 3 for naively calibrated and histogram-binned predictors under varying delay costs $\omega_0, \omega_1$ and information barrier $\theta$.

Many works have studied using machine learning to predict the onset of sepsis, as every hour of delayed treatment is associated with a 4-8% increase in mortality (Kumar et al., 2006; Reyna et al., 2020); existing works aim to better predict sepsis to treat high-priority patients earlier. Replicating results from Chicco & Jurman (2020) we train a binary predictor for sepsis onset using logistic regression on a dataset of 110,204 hospital admissions. The base predictor achieves an AUC of 0.86 using age, sex, and septic episodes as features. We then calibrate this predictor using both the naive method from Cho et al. (2022) (BINARY) and more nuanced histogram calibration (Zadrozny & Elkan, 2001) (CALIBRATED). Figure 3 shows the expected competitive ratio (normalized by the number of jobs $n = 100$) achieved by Algorithm 3 when provided advice from each of these predictors for varying delay costs $\omega_1, \omega_0$ and information barrier $\theta$. We see that the more nuanced predictions consistently result in schedules with smaller delay costs.

## 6. Conclusion

In this paper, we demonstrated that calibration is a powerful tool for algorithms with predictions in settings where performance is measured over a distribution and probabilistic estimates of a binary target enable good decisions. In particular, calibration bridges the gap between traditional theoretical approaches—which treat all predictions as equally reliable—and modern ML methodologies that offer fine-grained, instance-specific uncertainty quantification. We focused on the ski rental and online scheduling problems, developing online algorithms that exploit calibration guarantees to achieve strong average-case performance. For both problems, we highlighted settings where our algorithms outperform existing approaches and supported these findings with empirical evidence on real-world datasets.

This work exposes a number of directions for future research.

For ski rental, deriving performance guarantees in terms of binary cross entropy and focusing on less rigid calibration measures (e.g. expected calibration error) offer to further close the gap between theory and practice. More broadly, we believe calibration-based approaches offer broad potential for designing online decision-making algorithms beyond these two case studies, particularly in scenarios that require balancing worst-case robustness with reliable per-instance predictions.

## Impact Statement

This paper presents work whose goal is to advance the field of Machine Learning. There are many potential societal consequences of our work, none which we feel must be specifically highlighted here.

## Acknowledgments

This work was supported in part by NSF grant CCF-2338226, the Simons Foundation Collaboration on the Theory of Algorithmic Fairness, and a National Defense Science & Engineering Graduate (NDSEG) fellowship. We thank Bailey Flanigan for stimulating early discussions that inspired us to pursue this research direction, and Ziv Scully for a technical insight in the proof of Lemma 4.2.

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

## A. Ski Rental Proofs

**Lemma 3.2.** *Given a predictor $f$ with max calibration error $\alpha$, for all $v \in R(f)$,*

1. $\Pr[C \mid f(X) = v] \leq v + \alpha$

2. $\Pr[C^c \mid f(X) = v] \leq 1 - v + \alpha$

3. $\mathbb{E}[\mathrm{CR}(\mathcal{A}_k) \mid B_v, C] \leq 1 + \frac{k(v)}{b}$

4. $\mathbb{E}[\mathrm{CR}(\mathcal{A}_k) \mid B_v, C^c] \leq 1 + \frac{b \cdot \mathbb{1}_{\{k(v) < b\}}}{k(v)}$.

*Proof.* Recall that $B_v = \{f(X) = v\}$ is the event that $f$ predicts $v \in R(f)$, and $C = \{Z > b\}$ is the event that the true number of days skied is at least $b$. Because $f$ is a predictor of the indicator function $\mathbb{1}_C$ with max calibration error $\alpha$,

$$\Pr[C \mid B_v] = \Pr[Z > b \mid f(X) = v] = v - \alpha_v \leq v + \alpha$$

and

$$\Pr[C^c \mid B_v] = \Pr[Z \leq b \mid f(X) = v] = 1 - v + \alpha_v \leq 1 - v + \alpha.$$

This establishes (1) and (2). In the remainder of the proof we will reference the costs from conditions $(i)$-$(iv)$ in Table 3.

(3) $\mathbb{E}[CR(\mathcal{A}_k) \mid B_v, C] \leq 1 + \frac{k(v)}{b}$. Under the event $C$ ($Z > b$), one of conditions $(iii)$ or $(iv)$ must hold. The bound is tight when condition $(iv)$ holds. Under condition $(iii)$, it must be that $Z \leq k(v)$, so

$$\frac{\mathrm{ALG}(\mathcal{A}_k, f(X), Z)}{\mathrm{OPT}(Z)} = \frac{Z}{b} \leq \frac{k(v)}{b} \leq 1 + \frac{k(v)}{b}.$$

(4) $\mathbb{E}[CR(\mathcal{A}_k) \mid B_v, C^c] \leq 1 + \mathbb{1}_{\{k(v) < b\}} \cdot \frac{b}{k(v)}$.

Under the event $C^c$ ($Z \leq b$), one of conditions $(i)$ or $(ii)$ hold. The bound is trivial under condition $(i)$. Under condition $(ii)$, because $k(v) < Z$ and $Z \leq b$,

$$\frac{\mathrm{ALG}(\mathcal{A}_k, f(X), Z)}{\mathrm{OPT}(Z)} = \frac{k(v) + b}{Z} \leq \frac{k(v) + b}{k(v)} = 1 + \mathbb{1}_{\{k(v) < b\}} \cdot \frac{b}{k(v)}.$$

$\square$

**Theorem 3.3.** *Given a predictor $f$ with max calibration error $\alpha$, for any prediction $v \in R(f)$, Algorithm 1 achieves*

$$\mathbb{E}[\mathrm{CR}(\mathcal{A}_{k_*}) \mid f(X) = v] \leq$$
$$1 + 2\alpha + \min\left\{v + \alpha, 2\sqrt{(v + \alpha)(1 - v + \alpha)}\right\}.$$

*Proof.* Let $B_v = \{f(X) = v\}$ be the event that $f$ predicts $v \in R(f)$, and let $C = \{Z > b\}$ be the event that the true number of days skied is at least $b$. By the law of total expectation and Lemma 3.2,

$$\mathbb{E}[\mathrm{CR}(\mathcal{A}_k) \mid B_v] = \Pr[C \mid B_v] \cdot \mathbb{E}[\mathrm{CR}(\mathcal{A}_k) \mid C, B_v] + \Pr[C^c \mid B_v] \cdot \mathbb{E}[CR(\mathcal{A}_k) \mid C^c, B_v]$$
$$\leq (v + \alpha) \cdot \left(1 + \frac{k(v)}{b}\right) + (1 - v + \alpha) \cdot \left(1 + \mathbb{1}_{\{k(v) < b\}} \cdot \frac{b}{k(v)}\right)$$
$$= 1 + 2\alpha + \frac{(v + \alpha)k(v)}{b} + \mathbb{1}_{\{k(v) < b\}} \cdot \frac{(1 - v + \alpha)b}{k(v)}.$$

Finding the number of days to rent skis that minimizes this upper bound on competitive ratio amounts to solving two convex optimization problems — one for the case $k(v) < b$, and a second for $k(v) \geq b$ — then taking the minimizing solution.

(a) Minimize $\quad 1 + 2\alpha + \dfrac{(v + \alpha)\ell}{b} + \dfrac{(1 - v + \alpha)b}{\ell}$

$\qquad$ s.t. $\quad 0 \leq \ell \leq b$

(b) Minimize $\qquad 1 + 2\alpha + \dfrac{(v + \alpha)\ell}{b}$

$\qquad$ s.t. $\quad \ell \geq b$

Note first that (b) has optimal solution $\ell_* = b$. The Lagrangian of (a) is

$$\mathcal{L}(\ell, \lambda_1, \lambda_2) = 1 + 2\alpha + \frac{(v+\alpha)\ell}{b} + \frac{(1-v+\alpha)b}{\ell} + \lambda_1(\ell - b) - \lambda_2\ell$$

with KKT optimality conditions

$$\frac{v+\alpha}{b} - \frac{(1-v+\alpha)b}{\ell^2} + \lambda_1 - \lambda_2 = 0$$
$$\ell \leq b$$
$$-\ell \leq 0$$
$$\lambda_1, \lambda_2 \geq 0$$
$$\lambda_1(\ell - b) = 0$$
$$\lambda_2(-\ell) = 0.$$

We'll proceed by finding solutions to this system of equations via case analysis.

1. $\lambda_2 \neq 0$. Then, $\ell = 0$ and $\lambda_1 = 0$ by complementary slackness. But at least one of the stationarity or dual feasibility constraints are violated, since
$$0 > \frac{v+\alpha}{b} - \frac{(1-v+\alpha)b}{\ell^2} = \lambda_2.$$

2. $\lambda_2 = 0$ and $\lambda_1 \neq 0$. Then, $\ell = b$ by complementary slackness. Stationarity and dual feasibility are satisfied only when $0 \leq v \leq 0.5$, since in this case
$$\frac{v+\alpha}{b} - \frac{1-v+\alpha}{b} = -\lambda_1 \leq 0.$$

3. $\lambda_2 = 0$ and $\lambda_1 = 0$. Then, the first constraint gives that
$$\ell^2 = \frac{(1-v+\alpha)b^2}{v+\alpha}.$$

   Recall that $0 \leq \ell \leq b$, so this constraint is only satisfied when $0.5 \leq v \leq 1$ and $\ell = b\sqrt{\frac{1-v+\alpha}{v+\alpha}}$.

Because $\ell_* = b$ is the optimal solution to both (a) and (b) when $0 \leq v \leq 0.5$, it must be the case that $k_*(v) = b$ if $0 \leq v \leq 0.5$. When $0.5 < v \leq 1$, the optimal solution to (a) is $\ell_* = b\sqrt{\frac{1-v+\alpha}{v+\alpha}}$ and the optimal solution to (b) is $\ell_* = b$. The value of the former is $1 + 2\alpha + 2\sqrt{(v+\alpha)(1-v+\alpha)}$, while the value of the latter is $1 + 2\alpha + v + \alpha$. Taking the argmin yields

$$k_*(v) = \begin{cases} b & \text{if } 0 \leq v \leq \frac{4+3\alpha}{5} \\ b\sqrt{\frac{1-v+\alpha}{v+\alpha}} & \text{if } \frac{4+3\alpha}{5} < v \leq 1, \end{cases}$$

which is exactly Algorithm 1 and achieves a competitive ratio of

$$\mathbb{E}[\text{CR}(\mathcal{A}_{k_*}) \mid f(X) = v] \leq 1 + 2\alpha + \min\{v + \alpha, 2\sqrt{(v+\alpha)(1-v+\alpha)}\}.$$

$\square$

**Theorem 3.4.** *For all renting strategies* $k : [0,1] \to \mathbb{R}_+$*, predictions* $v \in [0,1]$ *and* $\epsilon > 0$*, there exists a distribution* $\mathcal{D}_v^\epsilon$ *and a calibrated predictor* $f$ *such that*

$$\mathbb{E}[\text{CR}(\mathcal{A}_k) \mid f(X) = v] \geq 1 + \min\left\{v, 2\sqrt{v(1-v)}\right\} - \epsilon.$$

*Table 3.* Objective values for fixed prediction $f(X) = v$, $z$ days skied, and renting for $k(v)$ days.

| CONDITION | OPT($z$) | ALG($\mathcal{A}_k, v, z$) |
|---|---|---|
| $(i)\;\; z \leq \min\{k(v),\, b\}$ | $z$ | $z$ |
| $(ii)\;\; k(v) < z \leq b$ | $z$ | $k(v) + b$ |
| $(iii)\;\; b < z \leq k(v)$ | $b$ | $z$ |
| $(iv)\;\; z > \max\{k(v),\, b\}$ | $b$ | $k(v) + b$ |

*Proof.* Let $v \in [0, 1]$ and $\epsilon > 0$. The calibrated predictor $f$ will deterministically output $v$, while the distribution $\mathcal{D}_v^\epsilon$ will depend on whether algorithm $\mathcal{A}_k$ buys before or after day $b$.

**Case 1:** $k(v) < b$. Define a distribution $\mathcal{D}_v^\epsilon$ where in a $v$ fraction of the data the true number of days skied is $z = b + \epsilon'$, and in a $1 - v$ fraction the number of days skied is $z = k(v) + \epsilon'$, where $\epsilon'$ is sufficiently small that

$$k(v) + \epsilon' \leq b \qquad \text{and} \qquad 2\sqrt{v(1-v)\left(1 - \frac{\epsilon'}{b}\right)} - \frac{2v\epsilon'}{b + \epsilon'} \geq 2\sqrt{v(1-v)} - \epsilon.$$

By construction, condition $(ii)$ from Table 3 is satisfied when $k(v) < b < z = b + \epsilon'$ with

$$\text{ALG}(\mathcal{A}_k, v, z)/\text{OPT}(z) = \frac{k(v) + b}{b + \epsilon'} = 1 + \frac{k(v) - \epsilon'}{b + \epsilon'}.$$

Similarly, condition $(ii)$ holds when $k(v) < z = k(v) + \epsilon' \leq b$ with

$$\text{ALG}(\mathcal{A}_k, v, z)/\text{OPT}(z) = \frac{k(v) + b}{k(v) + \epsilon'} = 1 + \frac{b - \epsilon'}{k(v) + \epsilon'}.$$

By the law of total expectation,

$$\mathbb{E}[CR(\mathcal{A}_k)] = v \cdot \left(1 + \frac{k(v) - \epsilon'}{b + \epsilon'}\right) + (1 - v) \cdot \left(1 + \frac{b - \epsilon'}{k(v) + \epsilon'}\right)$$

$$\geq \min_{\ell \geq 0} \left\{ v \cdot \left(1 + \frac{\ell - \epsilon'}{b + \epsilon'}\right) + (1 - v) \cdot \left(1 + \frac{b - \epsilon'}{\ell + \epsilon'}\right) \right\}.$$

Some basic calculus yields $\ell_* = \sqrt{\frac{1-v}{v}(b - \epsilon')(b + \epsilon')} - \epsilon'$, and evaluating the lower bound at $\ell^*$ gives

$$\mathbb{E}[CR(\mathcal{A}_k)] \geq 1 - \frac{2v\epsilon'}{b + \epsilon'} + 2\sqrt{v(1-v)\left(1 - \frac{\epsilon'}{b}\right)}$$

$$\geq 1 + 2\sqrt{v(1-v)} - \epsilon.$$

**Case 2:** $k(v) \geq b$.

Define a distribution $\mathcal{D}_v^\epsilon$ where in a $v$ fraction of the data the true number of days skied is $z = k(v) + \epsilon$, and in a $1 - v$ fraction the number of days skied is $z = b - \epsilon$. Condition $(iv)$ is satisfied when $b \leq k(v) < z = k(v) + \epsilon$ with $\text{ALG}(\mathcal{A}_k, v, z)/\text{OPT}(z) = 1 + \frac{k(v)}{b}$. Condition $(i)$ is satisfied when $z = b - \epsilon < b \leq k(v)$ with $\text{ALG}(\mathcal{A}_k, v, z)/\text{OPT}(z) = 1$. By the law of total expectation,

$$\mathbb{E}[CR(\mathcal{A}_k)] = v \cdot \left(1 + \frac{k(v)}{b}\right) + (1 - v) \cdot 1$$

$$\geq v \cdot 2 + (1 - v) \cdot 1$$

$$= 1 + v.$$

In both cases, $f$ is calibrated with respect to $\mathcal{D}_v^\epsilon$ since $\Pr[Z > b \mid f(X) = v] = v$. Moreover, because the cases are exhaustive, at least one of the corresponding lower bounds must hold. It follows immediately that

$$\mathbb{E}[CR(\mathcal{A}_k) \mid f(X) = v] \geq 1 + \min\left\{v, 2\sqrt{v(1-v)}\right\} - \epsilon.$$

$\square$

**Lemma 3.5.** *If $f : \mathcal{X} \to [0, 1]$ has mean-squared error $\eta$ and max calibration error $\alpha$, then*

$$\mathbb{E}[f(X)(1 - f(X))] \leq \eta + \alpha.$$

*Proof.* We have from the law of total expectation that

$$
\begin{aligned}
\eta &= \mathop{\mathbb{E}}_{(X,Z)\sim\mathcal{D}}\left[\left(\mathbb{1}_{\{Z>b\}} - f(X)\right)^2\right] \\
&= \sum_{v\in R(f)} \mathbb{E}\left[\left(\mathbb{1}_{\{Z>b\}} - v\right)^2 \,\Big|\, f(X) = v\right] \cdot \Pr[f(X) = v] \\
&= \sum_{v\in R(f)} \left(\mathbb{E}\left[\mathbb{1}_{\{Z>b\}} \mid f(X) = v\right] - 2v\,\mathbb{E}\left[\mathbb{1}_{\{Z>b\}} \mid f(X) = v\right] + v^2\right) \cdot \Pr[f(X) = v].
\end{aligned}
$$

Applying the definition of the local calibration error $\alpha_v$,

$$
\begin{aligned}
\eta &= \sum_{v\in R(f)} \left(\mathbb{E}\left[\mathbb{1}_{\{Z>b\}} \mid f(X) = v\right] - 2v\,\mathbb{E}\left[\mathbb{1}_{\{Z>b\}} \mid f(X) = v\right] + v^2\right) \cdot \Pr[f(X) = v] \\
&= \sum_{v\in R(f)} \left((v - \alpha_v) - 2v(v - \alpha_v) + v^2\right) \cdot \Pr[f(X) = v] \\
&= \sum_{v\in R(f)} \left(v(1 - v) + (2v - 1)\alpha_v\right) \cdot \Pr[f(X) = v] \\
&= \mathbb{E}[f(X)(1 - f(X))] + \sum_{v\in R(f)} (2v - 1)\alpha_v \cdot \Pr[f(X) = v].
\end{aligned}
$$

The observation that $(2v - 1)\alpha_v \geq -|\alpha_v|$ gives the result. $\square$

**Theorem 3.1.** *Given a predictor $f$ with mean-squared error $\eta$ and max calibration error $\alpha$, Algorithm 1 achieves $\mathbb{E}[CR(\mathcal{A}_{k_*})] \leq 1 + 2\alpha + \min\left\{\mathbb{E}[f(X)] + \alpha, 2\sqrt{\eta + 3\alpha}\right\}.$*

*Proof.* This result follows from Theorem 3.3, Lemma 3.5, and an application of Jensen's inequality. To begin,

$$
\begin{aligned}
\mathbb{E}[CR(\mathcal{A}_{k_*})] &= \mathbb{E}\left[\mathbb{E}[CR(\mathcal{A}_{k_*}) \mid f(X)]\right] && \text{(Tower property)} \\
&\leq \mathbb{E}\left[1 + 2\alpha + \min\left\{f(X) + \alpha, 2\sqrt{(f(X) + \alpha)(1 - f(X) + \alpha)}\right\}\right] && \text{(Theorem 3.3)} \\
&\leq 1 + 2\alpha + \min\left\{\mathbb{E}[f(X)] + \alpha, 2\,\mathbb{E}\left[\sqrt{(f(X) + \alpha)(1 - f(X) + \alpha)}\right]\right\},
\end{aligned}
$$

with the final line following from the fact that $\mathbb{E}[\min(X, Y)] \leq \min(\mathbb{E}[X], \mathbb{E}[Y])$ for random variables $X, Y$. Next, we argue from basic composition rules that the function $g(y) = \sqrt{(y + \alpha)(1 - y + \alpha)}$ is concave for $y \in [0, 1]$. The concavity of $g$ over its domain follows from the facts that (1) the $\sqrt{\cdot}$ function is concave and increasing in its argument and (2) $(y + \alpha)(1 - y + \alpha)$ is concave. Moreover, $g(y)$ is well-defined for all $y \in [0, 1]$. With concavity established, an application of Jensen's inequality yields

$$\mathbb{E}[CR(\mathcal{A}_{k_*})] \leq 1 + 2\alpha + \min\left\{\mathbb{E}[f(X)] + \alpha, 2\sqrt{\mathbb{E}[(f(X) + \alpha)(1 - f(X) + \alpha)]}\right\}.$$

To finish the proof, we will bound the term within the square root using Lemma 3.5. Notice that

$$(f(X) + \alpha)(1 - f(X) + \alpha) = f(X)(1 - f(X)) + \alpha + \alpha^2$$
$$\leq f(X)(1 - f(X)) + 2\alpha.$$

Finally,

$$\mathbb{E}[\mathrm{CR}(\mathcal{A}_{k_*})] \leq 1 + 2\alpha + \min\left\{\mathbb{E}[f(X)] + \alpha, 2\sqrt{\mathbb{E}[f(X)(1 - f(X))] + 2\alpha}\right\}$$
$$\leq 1 + 2\alpha + \min\left\{\mathbb{E}[f(X)] + \alpha, 2\sqrt{\eta + 3\alpha}\right\}. \qquad \text{(Lemma 3.5)}$$

$\square$

**Lemma 3.6.** *For all $a \in [0, 1/2]$, there exists an infinite family of input distributions for which Algorithm 2 defaults to a worst-case break-even strategy for all interval predictors $\mathrm{PIP}_\delta$ with uncertainty $\delta < a$.*

*Proof.* Let $a \in [0, 1/2]$ and consider a distribution that, for each unique feature vector $x \in \mathcal{X}$, has a true number of days skied that is either $z_1 \leq \frac{b}{2}$ with probability $1 - a$ or $z_2 \geq 2b$ with probability $a$. By construction, any interval prediction $\mathrm{PIP}_\delta(X) = [\ell, u]$ with $\delta < \min\{a, 1 - a\} = a$ must satisfy that $\ell \leq z_1$ and $u \geq z_2$. This means $b \in [\ell, u]$, so Algorithm 2 makes a determination of which day to buy based on the relative values of $\zeta(\delta, \ell)$, $\delta + \frac{u}{b}$, and 2. In particular, the algorithm follows the break-even strategy of buying on day $b$ when $\zeta(\delta, \ell) \geq 2$ and $\delta + \frac{u}{b} \geq 2$.

It is clear that $\delta + \frac{u}{b} \geq \frac{u}{b} \geq \frac{z_2}{b} \geq 2$. Next, recall the definition of $\zeta(\delta, \ell)$.

$$\zeta(\delta, \ell) = \begin{cases} \delta + (1 - \delta)\frac{b}{\ell} + 2\sqrt{\delta(1 - \delta)b/\ell} & \text{if } \delta \in [0, \frac{\ell}{b+\ell}) \\ 1 + \frac{b}{\ell} & \text{if } \delta \in [\frac{\ell}{b+\ell}, 1] \end{cases}$$

When $\delta \geq \frac{\ell}{b+\ell}$, we see that $\zeta(\delta, \ell) = 1 + \frac{b}{\ell} \geq 3$. To handle the case where $\delta < \frac{\ell}{b+\ell}$, we will show that

$$f(\delta, x) = \delta + (1 - \delta)x + 2\sqrt{\delta(1 - \delta)x} \geq 2$$

for all $x \geq 2$ and $\delta \in [0, 1/2]$. Plugging in $x = \frac{b}{\ell} \geq 2$ and noting that $\delta < \frac{\ell}{b+\ell} \leq 0.5$ implies the desired bound. Toward that end, notice that $f(\delta, x)$ is increasing in $x$, and so for all $x \geq 2$ we have that

$$f(\delta, x) \geq f(\delta, 2) = 2 - \delta + 2\sqrt{2\delta(1 - \delta)}.$$

All that is left is to show that $2\sqrt{2\delta(1 - \delta)} \geq \delta$. This is straightforward: for $\delta \in [0, 1/2]$,

$$2\sqrt{2(1 - \delta)} \geq \sqrt{1 - \delta} \geq \sqrt{\delta},$$

and multiplying through by $\sqrt{\delta}$ gives the desired inequality. In summary, we've shown that $b \in [\ell, u]$, $\zeta(\delta, \ell) \geq 2$, and $\delta + \frac{u}{b} \geq 2$ for the family of distributions described above. For this particular case, Algorithm 2 rents for $b$ days. $\square$

**Theorem 3.7.** *For all $a \in [0, 1/2]$, all instantiations $\mathcal{A}$ of Algorithm 2 using PIPs with uncertainty $\delta < a$, and all distributions from Lemma 3.6, if $f$ is a predictor with mean-squared error $\eta$ and max calibration error $\alpha$ satisfying $2\alpha + 2\sqrt{\eta + 3\alpha} < a$, then $\mathbb{E}[\mathrm{CR}(\mathcal{A}_{k_*})] < \mathbb{E}[\mathrm{CR}(\mathcal{A})]$.*

*Proof.* Let $a \in [0, 1/2]$ and consider any distribution from the infinite family given in Lemma 3.6. In particular, in any of these distributions, the number of days skied is greater than $b$ with probability $a$. Therefore, the expected competitive ratio of the break-even strategy that rents for $b$ days before buying is

$$\mathbb{E}[\mathrm{CR}(\mathcal{A})] = a \cdot 2 + (1 - a) \cdot 1 = 1 + a.$$

The result follows from the bound on $\mathbb{E}[\mathrm{CR}(\mathcal{A}_{k_*})]$ from Theorem 3.1. $\square$

# B. Scheduling Proofs

**Lemma 4.1.** *Let $X_1, \ldots, X_n$ be iid random variables with order statistics $X_{(1)} \geq \cdots \geq X_{(n)}$. For any symmetric function $g : \mathbb{R} \times \mathbb{R} \to \mathbb{R}$,*

$$\sum_{i=1}^{n} \sum_{j=i+1}^{n} g(X_{(i)}, X_{(j)}) = \sum_{i=1}^{n} \sum_{j=i+1}^{n} g(X_i, X_j).$$

*Proof.* Beginning with the facts that

$$\sum_{i=1}^{n} \sum_{j=1}^{n} g(X_{(i)}, X_{(j)}) = \sum_{i=1}^{n} \sum_{j=1}^{n} g(X_i, X_j) \quad \text{and} \quad \sum_{i=1}^{n} g(X_{(i)}, X_{(i)}) = \sum_{i=1}^{n} g(X_i, X_i),$$

it follows from the symmetry of $g$ that

$$\sum_{i=1}^{n} \sum_{j=i+1}^{n} g(X_{(i)}, X_{(j)}) = \frac{1}{2} \left( \sum_{i=1}^{n} \sum_{j=1}^{n} g(X_{(i)}, X_{(j)}) - \sum_{i=1}^{n} g(X_{(i)}, X_{(i)}) \right)$$

$$= \frac{1}{2} \left( \sum_{i=1}^{n} \sum_{j=1}^{n} g(X_i, X_j) - \sum_{i=1}^{n} g(X_i, X_i) \right)$$

$$= \sum_{i=1}^{n} \sum_{j=i+1}^{n} g(X_i, X_j).$$

$\square$

**Lemma 4.2.** *Let $X_1, \ldots, X_n$ be iid samples from a distribution over the unit interval $[0, 1]$ with order statistics $X_{(1)} \geq \cdots \geq X_{(n)}$. Then,*

$$\mathbb{E} \left[ \sum_{i=1}^{n} \sum_{j=i+1}^{n} (1 - X_{(i)}) \cdot X_{(j)} \right] \leq$$

$$\mathbb{E} \left[ \sum_{i=1}^{n} \sum_{j=i+1}^{n} (1 - X_i) \cdot X_j \right] - \binom{n}{2} \cdot \mathrm{Var}(X_1).$$

*Proof.* We'll begin by removing a shared term from both sides of the inequality. Notice that

$$\sum_{i=1}^{n} \sum_{j=i+1}^{n} X_{(i)} X_{(j)} = \sum_{i=1}^{n} \sum_{j=i+1}^{n} X_i X_j$$

by Lemma 4.1 with $g(x, y) = xy$. So, it is sufficient to show that

$$\mathbb{E} \left[ \sum_{i=1}^{n} \sum_{j=i+1}^{n} X_j \right] - \mathbb{E} \left[ \sum_{i=1}^{n} \sum_{j=i+1}^{n} X_{(j)} \right] \geq \binom{n}{2} \mathrm{Var}(X_1).$$

By linearity of expectation, the first term on the left-hand side is equal to $\binom{n}{2} \mathbb{E}[X_1]$. The random variables in the second term are not identically distributed, however, so a different approach is required. We will use a trick to express the sum in terms of the symmetric function $g(x, y) = \min(x, y)$, which allows us to remove the dependency on order statistics using Lemma 4.1.

$$\sum_{i=1}^{n} \sum_{j=i+1}^{n} X_{(j)} = \sum_{i=1}^{n} \sum_{j=i+1}^{n} \min\{X_{(i)}, X_{(j)}\} \qquad (X_{(i)} \geq X_{(j)} \text{ since } i \leq j)$$

$$= \sum_{i=1}^{n} \sum_{j=i+1}^{n} \min\{X_i, X_j\}. \qquad (\text{Lemma 4.1 with } g(x, y) = \min\{x, y\}])$$

Thus, the second term on the RHS is equal to $\binom{n}{2} \mathbb{E}[\min\{X_1, X_2\}]$. All that is left is to show that

$$\mathbb{E}[X_1] - \mathbb{E}[\min\{X_1, X_2\}] \geq \mathrm{Var}(X_1).$$

Toward that end, we can write

$$X_1 - \min\{X_1, X_2\} = \begin{cases} 0 & \text{if } X_1 \leq X_2 \\ |X_1 - X_2| & \text{if } X_1 > X_2, \end{cases}$$

so $\mathbb{E}[X_1] - \mathbb{E}[\min\{X_1, X_2\}] = \frac{1}{2}\mathbb{E}|X_1 - X_2|$. Finally, using the fact that $|X_1 - X_2| \in [0, 1]$ are iid, we have

$$\begin{aligned}
\frac{1}{2}\mathbb{E}|X_1 - X_2| &\geq \frac{1}{2}\mathbb{E}[(X_1 - X_2)^2] \\
&= \frac{1}{2}\mathbb{E}\left[X_1^2 - 2X_1 X_2 + X_2^2\right] \\
&= \frac{1}{2} \cdot 2\left(\mathbb{E}[X_1^2] - \mathbb{E}[X_1]^2\right) \\
&= \mathrm{Var}(X_1).
\end{aligned}$$

$\square$

**Theorem 4.3.** *Let $f$ be calibrated, with $\Pr[f(X) > \beta \mid Y = 0] = \epsilon_0$, $\Pr[f(X) \leq \beta \mid Y = 1] = \epsilon_1$,*

$$\kappa_1 = \Pr[f(X) > \beta]^2 \cdot \mathrm{Var}(f(X) \mid f(X) > \beta), \text{ and}$$
$$\kappa_2 = \Pr[f(X) \leq \beta]^2 \cdot \mathrm{Var}(f(X) \mid f(X) \leq \beta).$$

*Then*

1. $\mathbb{E}[L(f(\vec{X}), \vec{Y})] \leq \binom{n}{2}\left(\rho(1-\rho)(1+\epsilon_0)\epsilon_1 - \kappa_1\right)$
2. $\mathbb{E}[M(f(\vec{X}), \vec{Y})] = \binom{n}{2}(1-\rho)^2\epsilon_0^2$
3. $\mathbb{E}[N(f(\vec{X}), \vec{Y})] \leq \binom{n}{2}\left(\rho(1-\rho)\epsilon_0(1-\epsilon_1) - \kappa_2\right)$

*Proof.* We relax the calibration assumption and only assume that $f$ is monotonically calibrated, a weaker condition that the empirical frequencies $Z := \Pr[Y = 1 \mid f(X)]$ are non-decreasing in the prediction $f(X)$. Given $n$ jobs to schedule with features $\vec{X} = (X_1, \ldots, X_n)$ and the predictions $f(\vec{X}) = (f(X_1), \ldots, f(X_n))$, let $n_1 = |\{i : f(X_i) > \beta\}|$ be a random variable that counts the number of samples from $f$ with prediction larger than $\beta$, and define random variables $Z_i = \Pr[Y_i = 1 \mid f(X_i)]$ which give empirical frequencies. We'll begin by computing expectations conditioned on $n_1$ before taking an outer expectation.

$$\begin{aligned}
\mathbb{E}[L(f(\vec{X}), \vec{Y}) \mid n_1] &= \mathbb{E}\left[\mathbb{E}[L(f(\vec{X}), \vec{Y}) \mid f(\vec{X})] \mid n_1\right] && \text{(Tower property)} \\
&= \mathbb{E}\left[\mathbb{E}\left[\sum_{i=1}^{n_1}\sum_{j=i+1}^{n_1} \mathbb{1}_{\{\vec{Y}_{(i)}=0\}} \cdot \mathbb{1}_{\{\vec{Y}_{(j)}=1\}} \mid f(\vec{X})\right] \mid n_1\right] && \text{(Definition of } X) \\
&= \mathbb{E}\left[\sum_{i=1}^{n_1}\sum_{j=i+1}^{n_1} \Pr[\vec{Y}_{(i)} = 0 \mid f(\vec{X}_{(i)})] \cdot \Pr[\vec{Y}_{(j)} = 1 \mid f(\vec{X}_{(j)})] \mid n_1\right] && \text{(Independence)} \\
&= \mathbb{E}\left[\sum_{i=1}^{n_1}\sum_{j=i+1}^{n_1} (1 - Z_{(i)})Z_{(j)} \mid n_1\right].
\end{aligned}$$

Performing the same computation for counts $M(\cdot)$ and $N(\cdot)$ yields

$$\mathbb{E}[M(f(\vec{X}), \vec{Y}) \mid n_1] = \mathbb{E}\left[\sum_{i=1}^{n_1}\sum_{j=i+1}^{n_1} (1 - Z_{(i)}) \cdot (1 - Z_{(j)}) \mid n_1\right]$$

and

$$\mathbb{E}[N(f(\vec{X}), \vec{Y}) \mid n_1] = \mathbb{E}\left[\sum_{i=1}^{n_1} \sum_{j=n_1+1}^{n} (1 - Z_{(i)}) \cdot Z_{(j)} \mid n_1\right] + \mathbb{E}\left[\sum_{i=n_1+1}^{n} \sum_{j=i+1}^{n} (1 - Z_{(i)}) \cdot Z_{(j)} \mid n_1\right].$$

At this point, we can compute the conditional expectation of $M(\cdot)$ directly. By Lemma 4.1 with $g(x, y) = (1 - x)(1 - y)$,

$$\mathbb{E}\left[\sum_{i=1}^{n_1} \sum_{j=i+1}^{n_1} (1 - Z_{(i)}) \cdot (1 - Z_{(j)}) \mid n_1\right]$$

$$= \mathbb{E}\left[\sum_{i=1}^{n_1} \sum_{j=i+1}^{n_1} (1 - Z_i) \cdot (1 - Z_j) \mid n_1\right] \qquad \text{(Lemma 4.1)}$$

$$= \binom{n_1}{2} \cdot \mathbb{E}\big[\Pr[Y = 0 \mid f(X)] \mid f(X) > \beta\big]^2 \qquad \text{(Independence)}$$

$$= \binom{n_1}{2} \cdot \Pr[Y = 0 \mid f(X) > \beta]^2 \qquad \text{(Tower property)}$$

$$= \binom{n_1}{2} \cdot \frac{\epsilon_0^2 (1 - \rho)^2}{\Pr[f(X) > \beta]^2}. \qquad \text{(Bayes' rule)}$$

The same technique cannot be used to evaluate the expectations of $L(\cdot)$ and $N(\cdot)$ because the function $g(x, y) = (1 - x)y$ is not symmetric. Instead, we will provide upper bounds on the conditional expectations using Lemma 4.2, then evaluate the unordered results as before. For the conditional expectation of $L(\cdot)$, we have

$$\mathbb{E}\left[\sum_{i=1}^{n_1} \sum_{j=i+1}^{n_1} (1 - Z_{(i)}) \cdot Z_{(j)} \mid n_1\right]$$

$$\leq \mathbb{E}\left[\sum_{i=1}^{n_1} \sum_{j=i+1}^{n_1} (1 - Z_i) \cdot Z_j \mid n_1\right] - \binom{n_1}{2} \text{Var}(Z \mid f(X) > \beta) \qquad \text{(Lemma 4.2)}$$

$$= \binom{n_1}{2} \cdot \left(\Pr[Y = 0 \mid f(X) > \beta] \cdot \Pr[Y = 1 \mid f(X) > \beta] - \text{Var}(Z \mid f(X) > \beta)\right)$$

$$= \binom{n_1}{2} \cdot \left(\frac{\rho(1 - \rho)(1 - \epsilon_1)\epsilon_0}{\Pr[f(X) > \beta]^2} - \text{Var}(Z \mid f(X) > \beta)\right).$$

Similarly for the conditional expectation of the second term of $N(\cdot)$,

$$\mathbb{E}\left[\sum_{i=n_1+1}^{n} \sum_{j=i+1}^{n} (1 - Z_{(i)}) \cdot Z_{(j)} \mid n_1\right]$$

$$\leq \mathbb{E}\left[\sum_{i=n_1+1}^{n} \sum_{j=i+1}^{n} (1 - Z_i) \cdot Z_j \mid n_1\right] - \binom{n - n_1}{2} \text{Var}(Z \mid f(X) \leq \beta) \qquad \text{(Lemma 4.2)}$$

$$= \binom{n - n_1}{2} \cdot \left(\Pr[Y = 0 \mid f(X) \leq \beta] \cdot \Pr[Y = 1 \mid f(X) \leq \beta] - \text{Var}(Z \mid f(X) \leq \beta)\right)$$

$$= \binom{n - n_1}{2} \cdot \left(\frac{\rho(1 - \rho)(1 - \epsilon_0)\epsilon_1}{\Pr[f(X) \leq \beta]^2} - \text{Var}(Z \mid f(X) \leq \beta)\right).$$

For the first term of $N(\cdot)$, we simply apply the rearrangement inequality in lieu of Lemma 4.2 for unordering. Note that the sum has the form $\sum_i a_i \cdot b_i$, where $a_i = (1 - Z_{(i)})$ and $b_i = \sum_{j=i+1}^{n} Z_{(j)}$. The sequence $\{a_i\}_{i=1}^{n}$ is non-decreasing as a

result of the monotonic calibration of $f$, and $\{b_i\}_{i=1}^n$ is non-increasing. Thus,

$$\mathbb{E}\left[\sum_{i=1}^{n_1}\sum_{j=n_1+1}^{n}(1-Z_{(i)})\cdot Z_{(j)} \mid n_1\right] \leq \mathbb{E}\left[\sum_{i=1}^{n_1}\sum_{j=n_1+1}^{n}(1-Z_i)\cdot Z_j \mid n_1\right]$$

$$= n_1(n-n_1)\cdot \Pr[Y=0 \mid f(X)>\beta]\cdot \Pr[Y=1 \mid f(X)\leq\beta]$$

$$= n_1(n-n_1)\cdot \frac{\rho(1-\rho)\epsilon_1\epsilon_0}{\Pr[f(X)>\beta]\cdot \Pr[f(X)\leq\beta]}.$$

Next, we take an outer expectation to remove the dependency on $n_1$. Recall that $n_1$ follows a $\mathrm{Binomial}(n, \Pr[f(X)>\beta])$ distribution, so one can easily verify that

1. $\mathbb{E}[\binom{n_1}{2}] = \binom{n}{2}\cdot \Pr[f(X)>\beta]^2$

2. $\mathbb{E}[\binom{n-n_1}{2}] = \binom{n}{2}\cdot(1-\Pr[f(X)>\beta])^2 = \binom{n}{2}\cdot \Pr[f(X)\leq\beta]^2$

3. $\mathbb{E}[n_1(n-n_1)] = 2\binom{n}{2}\cdot \Pr[f(X)>\beta]\cdot \Pr[f(X)\leq\beta]$.

It follows immediately that

$$\mathbb{E}[L(f(\vec{X}),\vec{Y})] = \mathbb{E}[\mathbb{E}[L(f(\vec{X}),\vec{Y}) \mid n_1]]$$

$$\leq \mathbb{E}\left[\binom{n_1}{2}\cdot\left(\frac{\rho(1-\rho)(1-\epsilon_1)\epsilon_0}{\Pr[f(X)>\beta]^2} - \mathrm{Var}(Z \mid f(X)>\beta)\right)\right]$$

$$= \binom{n}{2}\cdot\left(\rho(1-\rho)(1-\epsilon_1)\epsilon_0 - \kappa_1\right)$$

$$\mathbb{E}[M(f(\vec{X}),\vec{Y})] = \mathbb{E}[\mathbb{E}[M(f(\vec{X}),\vec{Y}) \mid n_1]]$$

$$= \mathbb{E}\left[\binom{n_1}{2}\cdot\frac{\epsilon_0^2(1-\rho)^2}{\Pr[f(X)>\beta]^2}\right]$$

$$= \binom{n}{2}\cdot(1-\rho)^2\epsilon_0^2$$

$$\mathbb{E}[N(f(\vec{X}),\vec{Y})] = \mathbb{E}[\mathbb{E}[N(f(\vec{X}),\vec{Y}) \mid n_1]]$$

$$\leq \mathbb{E}\left[\binom{n-n_1}{2}\cdot\left(\frac{\rho(1-\rho)(1-\epsilon_0)\epsilon_1}{\Pr[f(X)\leq\beta]^2} - \mathrm{Var}(Z \mid f(X)\leq\beta)\right)\right]$$

$$+ \mathbb{E}\left[n_1(n-n_1)\cdot\frac{\rho(1-\rho)\epsilon_1\epsilon_0}{\Pr[f(X)>\beta]\cdot\Pr[f(X)\leq\beta]}\right]$$

$$= \binom{n}{2}\cdot\left(\rho(1-\rho)(1-\epsilon_0)\epsilon_1 + 2\rho(1-\rho)\epsilon_1\epsilon_0 - \kappa_2\right)$$

$$= \binom{n}{2}\cdot\left(\rho(1-\rho)(1+\epsilon_0)\epsilon_1 - \kappa_2\right),$$

where

$$\kappa_1 := \Pr[f(X)>\beta]^2\cdot\mathrm{Var}(Z \mid f(X)>\beta) \quad\text{and}\quad \kappa_2 := \Pr[f(X)\leq\beta]^2\cdot\mathrm{Var}(Z \mid f(X)\leq\beta).$$

The observation that $Z = f(X)$ when $f$ is calibrated gives the result from the main body. $\qquad\square$

## C. Experimental Details

### C.1. Ski-Rental: CitiBike

Our experiments with CitiBike use ridership duration data from June 2015. Although summer months have slightly longer rides, the overall shape of the distributions is similar across months (i.e. left-skewed distribution). Figure 4 illustrates the

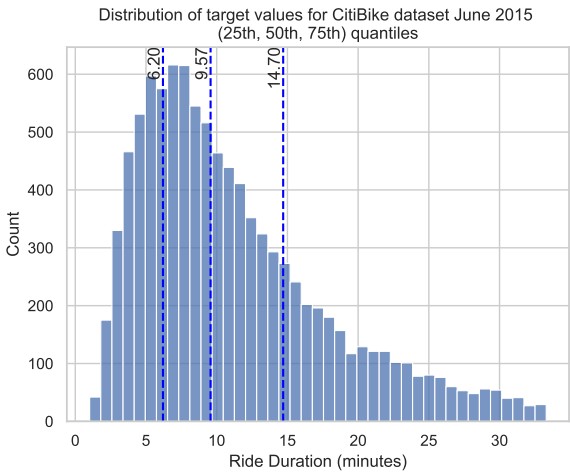

*Figure 4.* Distribution of ride times and quantiles in minutes, most rides are under 900 minutes.

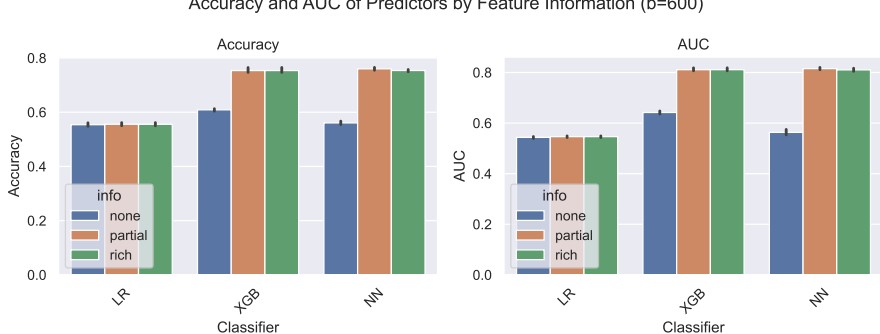

*Figure 5.* Predictor accuracy with different features around final docking station, no information, partial information (approximate latitude), and rich information (approximate latitude and longitude).

distribution of scores. This indicates that using this dataset for ski rental, the breakeven strategy will be better as $b$ increases since most of the rides will be less than $b$. This is an empirical consideration of running these algorithms that prior works do not consider. Thus, we select values of $b$ between 200 and 1000 as a reasonable interval for comparison.

**Feature Selection**   The original CitiBike features include per-trip features including user type, start and end times, location, station, gender, and birth year of the rider. We tested predictors with three types of feature: no information about final destination, partial information about final destination (end latitude only), and rich information about final destination (end longitude and latitude). Even with rich information, the best accuracy of the model's we consider are around 80% accuracy. This is because there are many factors affecting the ride duration. However with no information about the final destination, many of our models were close to random and thus do not serve as good predictor (Figure 5).

**Model Selection**   We tested a variety of models for both classification (e.g. linear regression, gradient boosting, XGBoost, k-Nearest Neighbors, Random Forest and a 2-layered Neural Network) and regression (e.g. Linear Regression, Bayesian Ridge Regression, XGBoost Regression, SGD Regressor, and Elastic Net, and 2-layered Neural Network). We ended up choosing three representative predictors of different model classes: regression, boosting, and neural networks. To fairly compare regression with classification we choose similar model classes: (Linear Regression, Logistic Regression), (XGBoost, XGBoost Regression), and two-layer neural networks.

**Calibration**   To calibrate an out-of-the box model, we tested histogram calibration (Zadrozny & Elkan, 2001), binned calibration (Gupta & Ramdas, 2021), and Platt scaling (Platt et al., 1999). While results from histogram and bin calibration

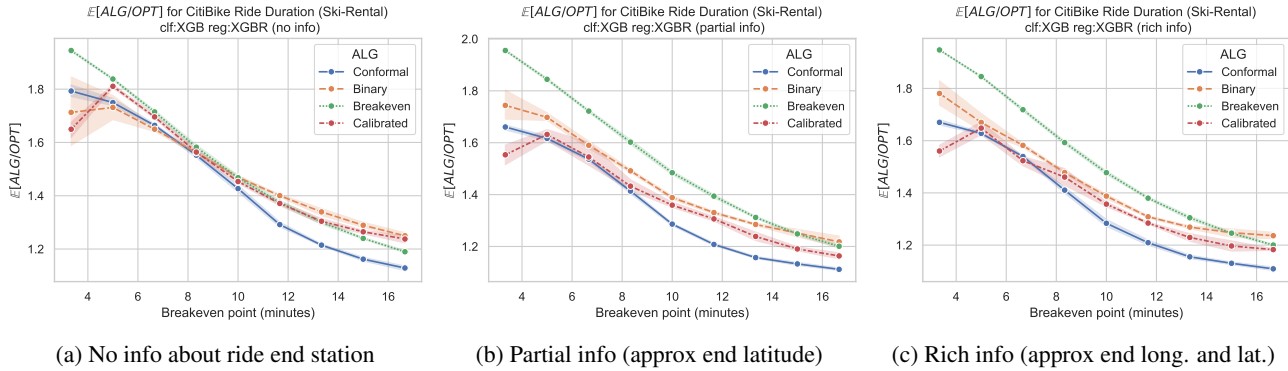

Figure 6. XGBoost predictors generally enable the calibrated predictor algorithm to do better than other baselines.

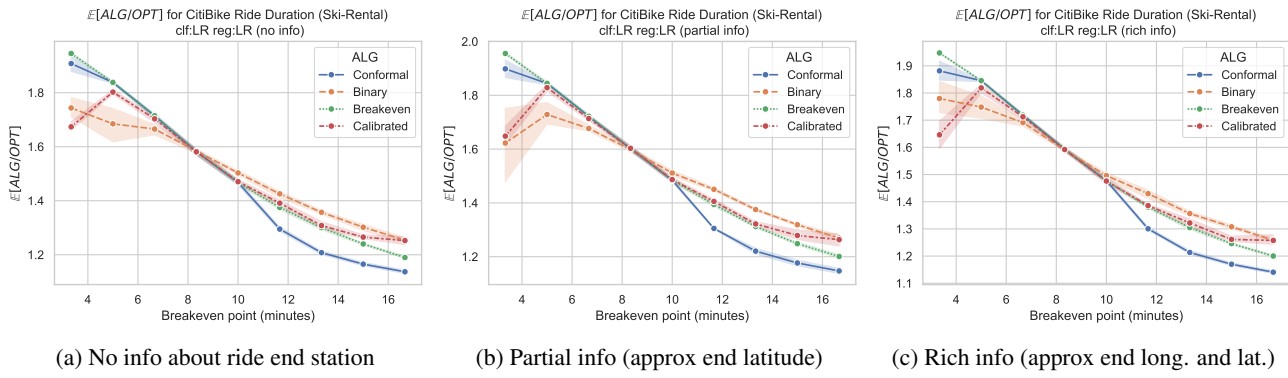

Figure 7. Linear regression and logistic regression remains similar to break even stretegy regardless of the features used.

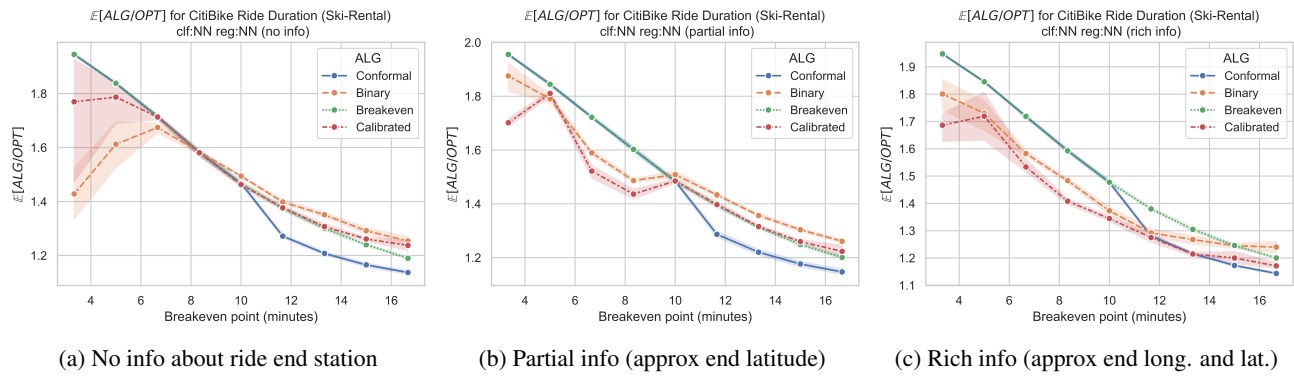

Figure 8. Neural network predictors generally enable calibrated predictor algorithm to do better than other baseline when there are informative features

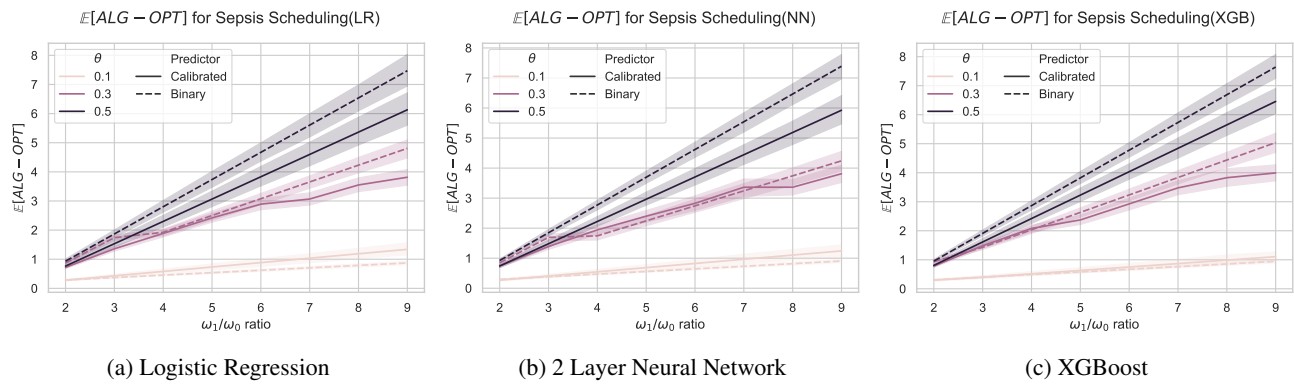

(a) Logistic Regression       (b) 2 Layer Neural Network       (c) XGBoost

*Figure 9.* Comparison of different base models. As $\theta$ increases the perforance of the calibrated preditctor becomes more similar to the binary predictor.

were similar, Platt scaling often produced calibrated probabilities within a very small interval. Though it is implemented in our code, we did not use it. A key intervention we make for calibration is to calibrate according to balanced classes in the validation set when the label distribution is highly skewed. This approach ensures that probabilities are not artificially skewed due to class imbalance.

**Regression** For a regression model as a fair comparison, we assume that the regression model also only has access to the 0/1 labels of the binary predictor for each $b$. To use convert the output conformal intervals to be used in the algorithm from Sun et al. (2024), we multiply the 0/1 intervals by $b$.

### C.2. Scheduling: Sepsis Triage

**Dataset** We use a dataset for sepsis prediction: 'Sepsis Survival Minimal Clinical Records'. [4] This dataset contains three characteristics: age, sex, and number of sepsis episodes. The target variable for prediction is patient mortality.

**Additional Models** We also include results for additional base models: 2 layer perception (Figure 9b) and XGBoost (Figure 9c)

---

[4] https://archive.ics.uci.edu/dataset/827/sepsis+survival+minimal+clinical+records

