# OpenReview forum: "Algorithms with Calibrated Machine Learning Predictions"
_ICML.cc/2025/Conference — ICML 2025 spotlightposter_

### Official Review · Reviewer_eD7y · 2025-03-08

**Overall Recommendation:** 4

**Summary:**

This paper introduces calibration as a tool to enhance learning-augmented algorithms, focusing on two problems: ski rental and online job scheduling. The authors propose algorithms that leverage calibrated predictions to achieve instance-specific competitive ratios, theoretically and empirically outperforming methods like conformal prediction. Experiments on real-world datasets (Citi Bike rentals and sepsis triage) validate their approach.

**Claims And Evidence:**

The claims made in the submission are supported by clear and convincing evidence.

**Essential References Not Discussed:**

All necessary related works are discussed in the submission.

**Experimental Designs Or Analyses:**

I only made a quick pass on experimental designs, and it looks reasonable. Note that the main contribution of this paper is theoretical.

**Methods And Evaluation Criteria:**

The proposed methods make sense for the problem.

**Other Comments Or Suggestions:**

n/a

**Other Strengths And Weaknesses:**

Strengths:

1. The paper proposes a new concept (i.e., calibrated prediction) for learning-augmented algorithms; this bridges the gap between global uncertainty assumptions in learning-augmented algorithms and modern ML models with local uncertainty estimates. Calibration is framed as a practical, instance-specific alternative to conformal prediction. This concept looks more practical compared to the existing prediction concepts.
2. From the theoretical view: for ski rental, the proposed algorithm achieves an expected competitive ratio bounded by calibration error and mean-squared error (Theorem 3.1). The authors also provide the lower bound (Theorem 3.4) shows near-optimality. For job Scheduling: By analyzing finer-grained calibrated predictors, the paper demonstrates reduced inversion costs (Theorem 4.3), improving upon prior work that relied on binary predictions.
3. This paper also includes the experimental results. They did experiments on real data (Citi Bike and sepsis) to demonstrate performance gains, linking theory to real-world applications.

Weakness:

There are some weaknesses in the results that were obtained. For ski rental, the ratio of the proposed algorithm (theorem 3.3.) is related to alpha, which is the calibrated error. This makes the algorithm lose its robustness. In the worst case, alpha can be very large. As the authors show in the lower bound theorem (theorem 3.4), the lower bound does not rely on alpha. For job scheduling, the obtained results rely on the iid assumption for features; this may not be true in reality.


Overall, although there are some weaknesses in the paper, I am still happy to see this paper in the proceeding since the paper proposes a new concept in learning-augmented algorithms, which might lead to some future works. Besides this, I also think that the theoretical understanding of this calibrated prediction concept would have a positive impact on reality. Thus, I recommend the acceptance.

**Questions For Authors:**

n/a

**Relation To Broader Scientific Literature:**

The key contribution of this paper is a new concept of the learning-augmented algorithm, i.e., using calibration to enhance the prediction. This is new and does not exist in the literature.

**Theoretical Claims:**

I checked the correctness of some claims but didn't go through all the proofs. The proofs that I have checked are reasonable and sound.

---

> ### Author Rebuttal · Authors · 2025-03-31
>
> We’re glad that the reviewer shares our excitement about calibration as a novel and practical prediction concept for algorithms with predictions with lots of theoretical potential!
>
> **“The expected performance bound for ski rental scales with max calibration error, which can be large, so the algorithm is non-robust.”**
> We thank the reviewer for the opportunity to clarify this point — while our global (Theorem 3.1) and prediction-level (Theorem 3.3) performance bounds degrade with larger calibration error, we note that the expected competitive ratio of our algorithm for ski rental will never exceed 3. This fact is not immediately obvious, and we will add an explanation to our discussion on worst-case expected performance in line 211 of Section 3.3 to help contextualize the 1.8 bound when calibration error is zero. The idea is that when the max calibration error is larger than ⅓, the algorithm executes a worst-case renting strategy that is 2-competitive. For max calibration error of at most ⅓, the bound from Theorem 3.3 can be no larger than 3.
>
> **“The results for job scheduling rely on an IID assumption that may not hold in reality”**
> Though prior work in ML-augmented scheduling (Cho et al., 2022) also assumes independence for the sake of simplifying the prediction task, we agree that features are likely to exhibit some level of correlation in real-world job scheduling contexts. The difficulty of the prediction task grows exponentially in the number of jobs when arbitrary correlations are allowed, but we see allowing for limited correlations as an interesting direction for future work to continue narrowing the gap between theory and practice.

---

### Official Review · Reviewer_wjo6 · 2025-03-09

**Overall Recommendation:** 3

**Summary:**

The paper studies how calibrated machine learning predictions can improve online algorithms. The paper focuses on two settings: the ski rental problem and online job scheduling.

In the ski rental problem, the predictions is about whether the skier will ski for more than a given threshold $b$ days using a calibrated binary predictor $f(X)$ where a calibrated $f(X)\in[0, 1]$ prediction means that it matches the actual likelihood of skiing more than $b$ days. The calibration error is the max calibration error, which measures the largest deviation from perfect calibration for any prediction.

In job scheduling, the model predicts the priority of each job using a calibrated probability score rather than a binary label, which allows for a finer-grained job ordering. Jobs are then scheduled using a threshold-based rule. In this setting, the authors consider the predictors to be perfectly calibrated.

In both settings, the authors show that the algorithm performance can be improved using calibrated predictors.

**Claims And Evidence:**

Yes

**Essential References Not Discussed:**

N/A

**Experimental Designs Or Analyses:**

Yes

**Methods And Evaluation Criteria:**

Yes

**Other Comments Or Suggestions:**

See above

**Other Strengths And Weaknesses:**

I like the idea of bringing the calibration to ensure the trustworthy property of provided predictions to the decision-makers. Unlike prior work that assumes global uncertainty of predictions, this approach uses instance-specific calibrated predictions, which can lead to more refined decision-making in the considered both ski rental and job scheduling problems. The authors also run a some experiments and it is good to see that the results indeed matches the theoretical findings.

Perhaps one of the weakness is the calibration metric considered in the paper. In the ski rent problem, the paper considers the max calibration error (i.e., a worst-case perspective), which sounds a bit weird to me, especially given that the authors have already assumed the underlying uncertainty is from distribution. One natural metric here should be expected calibration error, right?

Also, in the job schedule problem, the authors consider perfectly calibrated predictions, which is strong assumption. Could the results be generalized to a setting with predictors that may have certain calibration error here?

**Questions For Authors:**

See above

**Relation To Broader Scientific Literature:**

The paper may relates to broader literature in AI-assisted decision-making where AI provides predictions for decision-making tasks.

**Theoretical Claims:**

Yes

---

> ### Author Rebuttal · Authors · 2025-03-31
>
> We appreciate the reviewer’s feedback and are glad that they value our high-level goal of using calibration as a tool to ensure the trustworthiness of predictions provided to decision makers.
>
> **“Do the job scheduling results generalize to predictors with non-zero calibration error?”**
> Yes, our results extend beyond perfectly calibrated predictors. We presented our results in terms of perfectly calibrated predictions to highlight the intuition that finer-grained predictions reduce sequencing errors, but they generalize to any predictions that are monotonically calibrated. Monotonic calibration is the weaker condition that the empirical frequencies $\Pr(Y=1 \mid f(X))$ are non-decreasing in the prediction $f(X)$. This property holds trivially for perfectly calibrated predictors, but zero calibration error is not required. In fact, many calibration approaches used in practice (e.g. Platt scaling (Platt 1999) and isotonic regression (Zadrozny and Elkan 2002; 2003)) fit a monotonic function to data of the form $(f(X), \Pr(Y=1 \mid f(X))$, leading to a monotonically-calibrated predictor with non-zero calibration error.
>
> Thanks for pointing out that the statement of Theorem 4.3 appears brittle due to the assumption on perfect calibration. We’ll add a note that the result generalizes to predictors with non-zero calibration error and include full details for the more general case in the appendix.
>
> **“For ski rental, is an average error metric like expected calibration error (ECE) more thematically appropriate than a worst-case metric like max-calibration error (MCE)?”**
> Designing an algorithm that relies on ECE instead of MCE is an interesting direction for future work, and we will add it as an open question to Section 6. We suspect a global performance bound similar to Theorem 3.1 may still be recoverable in that setting. To the reviewer’s point, when MCE is significantly larger than ECE, Algorithm 1 makes conservative decisions, which suggests room for improvement. The fundamental challenge is that under ECE, prediction-specific performance bounds like those from Theorem 3.3 would only hold for predictions which are not “too miscalibrated,” rather than for all predictions.
> Nonetheless, MCE aligns with our goal of providing prediction-specific performance guarantees based on practical error metrics. Importantly, MCE is only worst-case with respect to the finite set of predictor outputs and not the distribution over instances. This means that MCE can be estimated from data, allowing for practical, data-driven guarantees.

---

### Official Review · Reviewer_Buyh · 2025-03-12

**Overall Recommendation:** 4

**Summary:**

The paper introduces a novel idea of leveraging calibrated predictors to design learning-augmented algorithms. Instead of proving consistency, robustness, and smoothness guarantees, which are worst-case guarantees, the authors derive bounds that depend on the predictor's maximum calibration error. They apply this approach to the ski-rental problem and a variant of the single-machine scheduling problem where all the jobs have unit sizes but different priorities. They also support their theoretical findings with extensive experiments (in Section 5 and Appendix C).

## update after rebuttal
The authors addressed most of my major concerns and questions during the rebuttal period. I am now convinced that the paper, after including the improvements discussed with the authors during rebuttal, would make a nice contribution to the literature of learning-augmented algorithms, and I have increased my score accordingly.

**Claims And Evidence:**

Yes

**Essential References Not Discussed:**

The setting studied in this paper appears related to algorithms with distributional predictions, where predictions correspond to probability distributions rather than specific realizations. However, the paper does not reference or compare with this line of work. Relevant papers include:
- "Binary Search with Distributional Predictions" (Dinitz et al., 2024)
- "Contract Scheduling with Distributional and Multiple Advice" (Angelopoulos, 2024)
- "Learning-Augmented Binary Search Trees" (Lin et al., 2022)

Additionally, in Section 3.3, the discussion on the tradeoff between worst-case guarantees (consistency/robustness) and the average performance is related to the paper "On Tradeoffs in Learning-Augmented Algorithms" (Benomar et al., 2025), which considers, for the ski-rental problem, a predictor of the form **1**$(z > b)$ that is accurate with probability q and studies trade-offs between consistency, robustness, smoothness, and the average-case performance of the algorithm.

**Experimental Designs Or Analyses:**

The experiments are well explained and their results are sound.

**Methods And Evaluation Criteria:**

Yes

**Other Comments Or Suggestions:**

Minor comment:
- L 144 (right column): $k$ is introduced but it is not yet defined at this point.

**Other Strengths And Weaknesses:**

The idea of using the calibration of the predictor to improve the bounds of learning-augmented algorithms is interesting and might inspire future work. However, unfortunately, the paper has many weaknesses, some of which can easily be addressed but others raise more serious concerns

**Inaccurate claims for motivating the setting:**
- The abstract mentions that the framework of learning-augmented algorithms "often
assumes uniform reliability across all predictions". This is inaccurate, or at least not clearly stated. The theoretical framework of learning-augmented algorithms makes no assumptions on the reliability of the predictions, and this is why its objective is to prove consistency and also robustness guarantees.
- (First page, first paragraph of the right column) The authors claim that most prior work focuses on extreme settings where predictions are either entirely accurate or completely uninformative. This is again not true. Most prior work also examines how algorithm performance scales with the sum of all errors, providing bounds that hold even when predictions are imperfect (smoothness). These results allow for performance guarantees when some predictions are accurate while others are not, and induce such bounds when the problem variables and predictions are stochastic.

**Unconventional/Inadequate notation and terminology**.
- The competitive ratio, as defined in the paper, is instance-dependent (it depends on the distribution D), whereas the standard definition refers to the worst-case ratio over all possible instances.
- The "additive competitive ratio" introduced in the paper is also instance-dependent and **is not even a ratio**. Calling it a competitive ratio is very inadequate and misleading. Moreover, it is not a suitable performance measure for the online scheduling problem (discussed further later).
- The notation ALG(A,I) to denote the output of algorithm A on instance I is unconventional. The standard notation would simply be A(I) (or ALG(I) if the algorithm is denoted ALG), as used for the optimal offline algorithm, denoted OPT(I), not ALG(OPT, I).
- The paper consistently uses the term "prediction-aided algorithms" instead of the standard term "learning-augmented algorithms".

**Inadequate performance measure: "additive competitive ratio"**.
In the online scheduling problem, what the authors refer to as a "competitive ratio" is actually a regret term E[ALG - OPT]. This terminology is very inadequate and misleading.
Standard performance evaluation for online algorithms typically uses the competitive ratio, defined as the worst-case ratio ALG/OPT, which provides a scale and size-independent comparison to the optimal solution.
It serves as a multiplicative approximation factor to the optimal solution. This is a standard approach in broader fields of algorithm design, where attaining optimal performance is either impossible or complex, and the aim becomes instead to have algorithms approximating the optimal output up to a multiplicative factor (NP-hard problems, heuristic algorithms,...).

Regret terms like E[ALG - OPT] are more common in online learning, where the goal is to analyze the rate at which ALG/OPT approaches 1. A more relevant regret measure in online algorithms would be E[ALG - CR * OPT].
Prior work on online scheduling in the learning-augmented setting (e.g., Cho et al., cited in the paper) evaluates algorithms using the standard competitive ratio.

The paper’s use of regret makes it difficult to compare results with existing literature and raises concerns about the relevance of the findings since regret analysis is not adapted for the considered setting.

**No robustness**
- The setting of the paper can be viewed as the standard learning-augmented setting with a different error measure, which is a combination of the calibration error and the instance-wise prediction error ($\eta$ in ski-rental). While their algorithms for both ski-rental and scheduling are 1-consistent, the authors do not prove any robustness bounds on their performance, that hold independently of the error (i.e. even if the error is arbitrarily large).

**Questions For Authors:**

- For the ski-rental problem, the paper considers a binary target variable and derives bounds based on the mean squared error (MSE) of the predictor $f$. However, in binary classification, binary cross-entropy is the default error measure and is commonly used for training calibrated classifiers. Can the authors justify why MSE is an appropriate choice in this context?
- MSE and calibration errors are presented as unrelated error measures. Are there any correlations/bounds between them? Are there any existing classifiers that minimize both the MSE and the max-calibration error? (i.e. is it indeed possible to have both $\alpha$ and $\eta$ close to 0)
- In Section 3.3., it is mentioned that the bound of Theorem 3.3 shows that the algorithm always has an expected competitive ratio of at most 1.8 when the classifier is calibrated. If the classifier is calibrated, i.e. $\alpha = 0$, then the bound becomes $1 + \min(\mathbb{E}[f(X)], 2\sqrt{\eta})$. Could the authors explain why this is always less than 1.8?

**Relation To Broader Scientific Literature:**

The paper contributes to the literature on learning-augmented algorithms. Instead of assuming that the predictions given to the algorithms can be arbitrary, the authors propose to leverage predictors with bounded calibration errors to prove new bounds, that improve upon prior works in some settings.

**Theoretical Claims:**

I read the proofs in the main paper (first 8 pages) and checked their correctness. I did not read the proofs in the appendices, but the proof sketches explained in the paper seem correct and convincing.

---

> ### Author Rebuttal · Authors · 2025-03-31
>
> We sincerely thank the reviewer for their detailed feedback. However, we believe there may have been a significant misunderstanding regarding key aspects of our paper. Below, we address each concern in detail and respectfully ask that the reviewer reconsider their evaluation based on these clarifications and feedback from other reviewers.
>
> **Motivation of the setting.**
> We believe the reviewer’s concerns stem from a misunderstanding of our motivation for calibrated predictions:
> - When stating that learning-augmented algorithms “often assume uniform reliability across all predictions” in the abstract, our intention was to highlight a reliance on global parameters that reflect a user's trust in the predictions in aggregate (e.g., often denoted $\lambda \in [0,1]$ in prior work, with $\lambda = 1$ indicating no trust and $\lambda = 0$ full trust). Our approach uses prediction-level uncertainty from calibration to eliminate the need for parameter tuning. We will revise line 015 of the abstract accordingly. Reviewer eD7y recognized this distinction, stating "this paper bridges the gap between global uncertainty assumptions in learning-augmented algorithms and modern ML models with local uncertainty estimates."
> - The reference to “extreme settings” on page 1 was meant only to illustrate the endpoints of the global uncertainty parameter, not to imply that existing work studies these extremes. We will clarify this in line 015, column 2 to avoid ambiguity.
>
>
> **Essential references.**
> We appreciate the reviewer’s insightful connection to distributional predictions. We will cite the suggested references (and “Learning Online Algorithms with Distribution Advice” (Diakonikolas et al., 2021)) in Section 2.1, noting that they are conceptually related but do not study uncertainty quantification.
>
> Regarding the **concurrent work** (posted on arXiv on 1/22/25) by Benomar et al. (2025), their approach gives average-case bounds for ski rental assuming access to a predictor that correctly guesses $I(Z > b)$ with known probability for any input. This type of assumption — often called “conditional coverage” in the ML literature (see, e.g., “Distribution-free Prediction Bands for Nonparametric Regression” (Lei and Wasserman 2012)) — enables better bounds, but is much stronger than our calibration assumptions.
>
> **Performance measures.**
> We strongly disagree that our performance measures are inadequate.
> - As Reviewer v9Cf stated, “in theory, the expected competitive ratio is the most common objective and evaluation criterion for these types of algorithms,” supporting our methodology. Indeed, the expected competitive ratio we employ has precedent in sample-based learning-augmented algorithms (e.g., Diakonikolas et al., 2021; Anand et al., 2022), which align closely with our approach.
> - We will update our terminology from “additive competitive ratio” to “regret,” as suggested. Many prior works on learning-augmented scheduling algorithms employ regret, with results given as explicit additive bounds (e.g., “Learning-Augmented Algorithms with Explicit Predictors” (Elias et al., 2024); “Non-clairvoyant Scheduling with Predictions (Im et al., 2023)), or later converted to competitive ratios (e.g., “Permutation Predictions for Non-Clairvoyant Scheduling” (Lindermayr and Megow 2022). Though Cho et al. (2022) follow the second approach, bounding expected regret is a core component of their analysis.
> We will add this context to the third paragraph of Section 2.
>
> **Robustness.**
> We agree that our paper differs from traditional robustness analyses. Our primary objective is bounding expected performance degradation based on calibration error (CE), which captures robustness in our distributional setting. We will clarify this after the definition of CE in Section 2. Please see the responses to Reviewers wjo6 and eD7y for discussion on performance guarantees when CE may be large.
>
> **“How are MSE and CE related, and can both be 0?”**
> A classic result from “On Subjective Probability Forecasting” (Saunders 1963) decomposes MSE into a sum of non-negative refinement and calibration terms. This means $MSE=0$ implies $CE=0$. We will note this after the definition of CE in Section 2.
>
> **“Why MSE and not binary cross entropy (BCE)?”**
> We use MSE because its refinement term arises naturally in our analysis, and its calibration term is a standard error measure. While a similar decomposition holds for BCE, its refinement and calibration terms are not well-suited to our setting. We thank the reviewer for highlighting this mismatch between the error metrics used to train an ML model vs provide performance guarantees; it is a gap that remains between theory and practice, and an interesting question for future work that we will add to Section 6.
>
> **“Why is the expected CR at most 1.8 when CE is zero?”** When $\alpha=0$, the prediction-level upper bound from Theorem 3.3 achieves a maximum value of 1.8 at $v=0.8$. We will add this brief explanation to Section 3.3.

---

> > ### Comment · Reviewer_Buyh · 2025-04-02
> >
> > I thank the authors for their detailed response.
> >
> > **Competitive ratio.**
> > Regarding performance measures, the competitive ratio is indeed the standard measure for analyzing online algorithms, and I did not claim otherwise. My point was that the competitive ratio is defined as the **worst-case** ratio over all possible instances, whereas the definition you consider in the current paper is instance-dependent (it depends on the distribution $D$). For example, in "Learning Online Algorithms with Distributional Advice" (Diakonikolas et al.), which you cited in your response, the authors establish **worst-case** guarantees on the ratio $\text{cost}(A, D)/\text{OPT}_D$ over all distributions $D$ in a given class $\mathcal{C}$, independent of the specific distribution.
> >
> > **Regret.**
> > Regarding the regret analysis, I agree that using it is reasonable. I guess my main concern was with the terminology: referring to it as an "additive competitive ratio" was highly misleading.
> >
> > ### **Robustness**
> > The major remaining weakness is the lack of robustness guarantees. As I noted in my review, while the authors introduce calibration error as their chosen error measure, which is well-motivated, it does not replace the need for robustness guarantees.
> >
> > - For the ski-rental problem, the authors stated in their response to reviewer eD7y that the ratio $\text{ALG}/\text{OPT}$ is always at most 3. However, this holds only because their algorithm assumes knowledge of the max-calibration error $\alpha$. In the context of learning-augmented algorithms, an algorithm is considered robust if it can still perform reasonably well without any knowledge of prediction quality. This is why I am not convinced by the robustness claimed in the authors response.
> > - Additionally, in prior work, if the maximum prediction error (under any given error measure) is known, the parameter $\lambda$ can be chosen optimally, eliminating the need for tuning it. Thus I also don't agree with the claim that the approach of the paper eliminates the need for tuning the levels of robustness and consistency.
> > - The same issues arises in the scheduling problem, where no robustness bounds are provided. The authors’ explain in their response that their bounds capture robustness based on calibration error. However, error-dependent bounds, which indeed ensure bounded performance degradation for bounded error, indicate "smoothness" rather than "robustness". Robustness, on the other hand, requires guarantees that hold even under arbitrary prediction errors.
> >
> > ### **Other comments**
> > I am satisfied with the authors’ responses to my other comments and questions, and encourage them to include the corresponding discussions in the paper.
> >
> > I have slightly raised my score, but I still do not recommend acceptance, as robustness remains a fundamental requirement (alongside consistency and smoothness) when designing algorithms with predictions. I would be happy to reconsider my score if the authors address this concern during the discussion period.

---

> > > ### Author Response · Authors · 2025-04-08
> > >
> > > We thank the reviewer for their continued engagement and willingness to reconsider their evaluation. After reflecting carefully on the latest comments, we believe it is possible to address the remaining concerns raised regarding robustness. We’re grateful for this suggestion, which we agree will strengthen the paper, and hope these clarifications will encourage the reviewer to raise their recommendation into the acceptance range.
> > >
> > > To begin, we briefly reiterate the core motivation of our paper: addressing critical gaps between theory and practice in algorithms-with-predictions. A significant gap is the widespread reliance on the maximum prediction error in prior work—a worst-case metric that is inherently impossible to estimate reliably. To clarify the distinction in our setting:
> > >
> > > - **The max calibration error $\\alpha$ of a ML predictor is estimable**. As noted in our response to Reviewer wj06, $\alpha$ is defined as a maximum expected calibration deviation over the finite set of predictor outputs. Importantly, this quantity is not worst-case with respect to the underlying input space. In fact, an estimate of $\\alpha$ is a natural byproduct of post-hoc calibration procedures used in practice.
> > > - In contrast, **the maximum prediction error of a ML predictor—as used in prior work—is not estimable given finite samples**, since it is defined as a supremum over the input space. To illustrate, consider two distributions that differ only in a single point label (one label 0 and the other labeled 1, for example). If the support of the distributions is continuous or large, no ML model can statistically distinguish between these distributions given finite samples. Thus, for any ML model, there are distributions where its maximum prediction error is arbitrarily large.
> > >
> > > In light of the fact that knowledge of $\\alpha$ is a reasonable assumption in practice, the primary focus of our paper is strong average-case performance when given (1) query access to the ML predictor and (2) additional (attainable) calibration metrics. The second requirement clearly differentiates our approach from prior work. Nevertheless, directly addressing the reviewer's concern, we have found that it is indeed possible to incorporate worst-case robustness guarantees, which we will add to the final version of the paper:
> > >
> > > - For ski rental, an analysis similar to that of Theorem 15 in Anand et al. (2020) shows that Algorithm 1 is $g(\alpha)$-robust, where $$g(\\alpha) = \\begin{cases} 1 + \sqrt{\\frac{1+\\alpha}{\\alpha}} &\text{if } \\alpha \\in [0, \\frac{1}{3}) \\\\ 2 &\text{if } \alpha \\in [\\frac{1}{3}, 1] \\end{cases}$$  (a decreasing function of $\\alpha$). This is because Algorithm 1 executes a worst-case 2-competitive strategy when $\\alpha \\geq \\frac{1}{3}$, and as noted in line 140 column 2, Algorithm 1 never buys skis before day $b\\sqrt{\\frac{\\alpha}{1+\alpha}}$ for $\\alpha < \\frac{1}{3}$. If $g(\alpha)$ is larger than some desired robustness threshold $\\beta$, one can artificially increase $\\alpha$, running the algorithm using a max calibration error bound of $\\alpha’ > \\alpha$ such that $g(\\alpha’) < \\beta$. As seen from the expected performance bounds in Theorems 3.1 and 3.3, this adjustment will come at the cost of average-case performance, highlighting the tradeoff between average and worst-case performance.
> > > - For scheduling, we generalize the approach of Cho et al. (2022) by allowing for predictions from an arbitrary calibrated predictor. The error metric in this setting is not calibration error (see the response to Reviewer wjo6 about how we only require monotonic calibration), but rather the generalized false negative/false positive rates $\\epsilon_0$ and $\\epsilon_1$, which represent the discriminative power of the predictor. Since we recover the regret bounds from Cho et al. (2022), our approach inherits their robustness guarantees, which hold when $\\epsilon_0 = \\epsilon_1 = \\frac{1}{2}$, i.e., the predictions are truly random.
> > >
> > > By justifying knowledge of the max calibration error $\\alpha$ in practice and giving explicit robustness guarantees, we directly address the reviewer’s remaining concerns and illustrate the fundamental tradeoff between robustness and average-case performance. We thank the reviewer again for their suggestions and hope these clarifications fully resolve the issue.

---

### Official Review · Reviewer_v9Cf · 2025-03-19

**Overall Recommendation:** 4

**Summary:**

The paper initiates the study of the effect of calibration in algorithms with predictions through two case studies:
1. Ski Rental: The authors design an algorithm that achieves optimal prediction-dependent performance, bounding the expected performance using both the squared error and the calibration error. They proved that calibrated predictions can outperform the conformal prediction method for infinitely many instances.
2. Online Job Scheduling: They prove that a properly calibrated predictor with finer-grained confidence levels yields better performance bounds than prior work.
The theoretical findings are supported by experiments on real-world datasets, showing that the proposed algorithms leveraging calibrated predictions outperform the baselines.

**Claims And Evidence:**

Most of the claims in the paper are convincing, but I have two concerns:
1. Clarification on Line 347-348 (Right-Hand Side): When you state that the bound is "tight," does this mean that there is a matching lower bound, or does it imply that the upper bound is optimal? A clearer explanation of what "tight" means in this context would be helpful.
2. Comparison with Conformal Prediction Methods: The claim that "there exist infinitely many families of distributions where calibrated predictions outperform conformal prediction methods" does not necessarily mean superiority. To make a stronger argument, you would need to show that the inverse does not hold—i.e., that there are no such infinite families where conformal prediction consistently outperforms calibrated predictions. Additionally, since the bound includes the squared loss, it is unclear whether calibration itself is responsible for the improved performance. A predictor with high squared error, even if well-calibrated, might still yield a poor bound. Authors might need to be careful to make a strong claim.

**Essential References Not Discussed:**

To the best of my knowledge, the paper appropriately cites all essential related works.

**Experimental Designs Or Analyses:**

The experimental design appears to be valid and well-structured. However, based on the figures, I am not fully convinced that the algorithm with calibrated predictors consistently outperforms the conformal prediction baseline.

**Methods And Evaluation Criteria:**

Yes, in theory, the expected competitive ratio is the most common objective and evaluation criterion for these types of algorithms. Additionally, the chosen dataset is appropriate and aligns well with the study's objectives.

**Other Comments Or Suggestions:**

1. Line 779: Did you define K_1(f,D) in the paper?

**Other Strengths And Weaknesses:**

Strengths:
1. The paper is innovative, being the first to study the effect of calibration in algorithms with predictions. The proposed algorithms and analysis are novel and interesting.
2. The writing is clear and well-structured.

Weaknesses:
1. The techniques and results for the two case studies feel somewhat disconnected. It remains unclear, at a high level, why and when calibration is beneficial for algorithms with predictions. A more detailed explanation and discussion would strengthen the paper.
2. The experimental results are not entirely convincing. As mentioned earlier, it is unclear whether the algorithm with calibrated predictors consistently outperforms the conformal prediction baseline. Additionally, it would be valuable to include experiments that vary the calibration error (while controlling for squared loss, if possible) to empirically demonstrate how calibration affects performance.

**Questions For Authors:**

Could you clarify the conclusion that the calibration-based algorithm outperforms the conformal prediction method? The theoretical results suggest that the conformal prediction approach fails in certain cases, but this does not necessarily imply that the calibration-based method is strictly superior in general. Could you provide additional theoretical justification to support this claim?

Additionally, I would appreciate further clarification on the concerns raised earlier.

**Relation To Broader Scientific Literature:**

The paper initiate the study of calibrated predictors for algorithms with predictions, and provided interesting theoretical analysis for two case studies.

**Theoretical Claims:**

I checked the proofs in Section 3 and they seem to be correct. For Section 4, I briefly reviewed the proofs but did not check the details in depth.

---

> ### Author Rebuttal · Authors · 2025-03-31
>
> We thank the reviewer for their thoughtful feedback! We’re encouraged that they find our application of calibration to algorithms with predictions innovative, and our algorithms and analysis novel and interesting. In addition to the modifications detailed below, we will correct the typo on line 776 for the camera-ready version of the paper.
>
> **“Does the calibration-based algorithm outperform the conformal prediction method for ski rental?”**
> We see the calibration-based and conformal prediction-based algorithms as two orthogonal approaches with neither being strictly superior. We solely aim to point out (e.g. in lines 42 column 2, 78 column 1, 244 column 1, and Theorem 3.7) that calibration offers advantages in situations where conformal predictors struggle: (1) when only binary targets are available for training an ML model, or (2) there is high variance in the target that is unexplained by the features. In general, the better method will depend both on the underlying ML model and the distribution over features and instances. This can be seen in the ablations for different models in the appendix, and we will modify the conclusion from lines 427-428 column 1 to better communicate this fact.
>
> **“Lines 347-348 column 2: what does it mean that the bound is tight?”**
> Thank you for pointing out this imprecise wording. It will be replaced by “the inequalities hold with equality.”
>
> **“Why and when is calibration beneficial for algorithms with predictions?”**
> This is a great question. Calibration is beneficial for algorithms with predictions in settings where (1) the goal is good performance over a distribution of instances, (2) there is a binary quantity of interest that is predictable, and (3) probabilistic estimates of this quantity are sufficient to make good decisions. Both case studies we consider fall under this framework. In ski rental, we provide a principled approach for decision making given a probabilistic estimate of whether a skier will ski for at least b days. For job scheduling, there is an optimal scheduling strategy based on the probabilities that each job is high priority.
>
> To see why calibration is beneficial, notice that in a Bayesian sense, the best possible probabilistic estimate of a binary quantity $Y$ given any (potentially untrusted) prediction $f(X)$ of $Y$ is $\Pr(Y=1 \mid f(X))$. Computing these probabilities corresponds exactly to calibrating the original predictor $f$ over the input distribution, but this post-hoc correction is not always possible (for example, if the decision-maker does not have access to the full ML model and data). As a result, (approximate) calibration is a precondition for the decision-maker being able to generate reliable probabilistic estimates of the quantity of interest. If there are no calibration guarantees, we enter the classic realm of algorithms-with-predictions, where algorithms need to be robust against arbitrarily error-prone predictions. This is the reason calibrated predictions are often referred to as “trustworthy.” We agree that a discussion of this kind in Section 6 will help to unify and strengthen the paper, and we thank the reviewer for this recommendation.
>
> **“Can experiments vary calibration error while controlling for squared loss?”**
> Because calibration error is an additive component of the mean-squared error — see the response to Reviewer Buyh on their relationship for more details — varying calibration error necessarily changes the mean-squared error. This makes it difficult to design an experiment, such as the one that was suggested, that isolates the effects of calibration error vs overall predictor efficacy.

---

> > ### Comment · Reviewer_v9Cf · 2025-04-08
> >
> > I would like to thank the reviewers for answering my questions. I will keep my score.

---

### Decision · Program_Chairs · 2025-05-01

**Decision:**

Accept (spotlight poster)

**Comment:**

This paper introduces calibrated predictions as a refined way to incorporate machine learning predictions into classical online algorithms, studied through two canonical problems: ski rental and single-machine scheduling. By focusing on calibration, the paper aims to provide instance-specific performance guarantees, thereby moving beyond global, worst-case error assumptions typically used in learning-augmented algorithms. The authors prove that when the maximum calibration error is sufficiently small, their algorithms achieve performance bounds that can, in certain instances, improve upon methods based on conformal prediction or classical global-error metrics. Extensive theoretical analysis and real-data experiments illustrate the practical potential of leveraging calibration in online decision-making.

After careful deliberation, the consensus is to accept this paper. The reviewers found the idea of leveraging calibration in learning-augmented algorithms to be both novel and practically relevant, addressing gaps in current approaches that rely on globally bounded errors. The authors’ detailed rebuttal and discussion resolved most concerns, especially around worst-case performance and the scope of the scheduling results. Collectively, the reviewers felt that the paper’s contributions, both conceptual and theoretical, justify publication at ICML.